# RESIDUAL DENOISING DIFFUSION MODELS

## ABSTRACT

We propose residual denoising diffusion models (RDDM), a novel dual diffusion process that decouples the traditional single denoising diffusion process into residual diffusion and noise diffusion. This dual diffusion framework expands the denoising-based diffusion models, initially uninterpretable for image restoration, into a unified and interpretable model for both image generation and restoration by introducing residuals. Specifically, our residual diffusion represents directional diffusion from the target image to the degraded input image and explicitly guides the reverse generation process for image restoration, while noise diffusion represents random perturbations in the diffusion process. The residual prioritizes certainty, while the noise emphasizes diversity, enabling RDDM to effectively unify tasks with varying certainty or diversity requirements, such as image generation and restoration. We demonstrate that our sampling process is consistent with that of DDPM and DDIM through coefficient transformation, and propose a partially path-independent generation process to better understand the reverse process. Notably, our RDDM enables a generic UNet, trained with only an $\ell_1$ loss and a batch size of 1, to compete with state-of-the-art image restoration methods. We provide code and pre-trained models to encourage further exploration, application, and development of our innovative framework.

## 1 INTRODUCTION

In real-life scenarios, diffusion often occurs in complex forms involving multiple, concurrent processes, such as the dispersion of multiple gases or the propagation of different types of waves or fields. This leads us to ponder whether the denoising-based diffusion models (Ho et al., 2020; Song et al., 2021a) have limitations in focusing solely on denoising. Current diffusion-based image restoration methods (Lugmayr et al., 2022; Saharia et al., 2022; Rombach et al., 2022; Jin et al., 2022b; Özdenizci & Legenstein, 2023) extend the diffusion model to image restoration tasks by using degraded images as a condition input to implicitly guide the reverse generation process, without modifying the original denoising diffusion process (Ho et al., 2020; Song et al., 2021a). However, the reverse process starting from noise seems to be unnecessary, as the degraded image is already known. The forward process is non-interpretability for image restoration, as the diffusion process does not contain any information about the degraded image.

In this paper, we explore a novel dual diffusion process and propose Residual Denoising Diffusion Models (RDDM), which can tackle the non-interpretability of a single denoising process for image restoration. In RDDM, we decouple the previous diffusion process into residual diffusion and noise diffusion. Residual diffusion prioritizes certainty and represents a directional diffusion from the target image to the conditional input image, and noise diffusion emphasizes diversity and represents random perturbations in the diffusion process. Thus, our RDDM can unify different tasks that require different certainty or diversity, e.g., image generation and restoration. Compared to denoising-based diffusion models for image restoration, the residuals in RDDM clearly indicate the forward diffusion direction and explicitly guide the reverse generation process for image restoration.

Specifically, we redefine a new forward process that allows simultaneous diffusion of residuals and noise, wherein the target image progressively diffuses into a purely noisy image for image generation or a noise-carrying input image for image restoration, as shown in Fig. 1. Unlike the previous denoising diffusion model (Ho et al., 2020; Song et al., 2021a), which uses one coefficient schedule to control the mixing ratio of noise and images, our RDDM employs two independent coefficient schedules to control the diffusion speed of residuals and noise. We found that this independent diffusion property is also evident in the reverse generation process, e.g., readjusting the coefficient

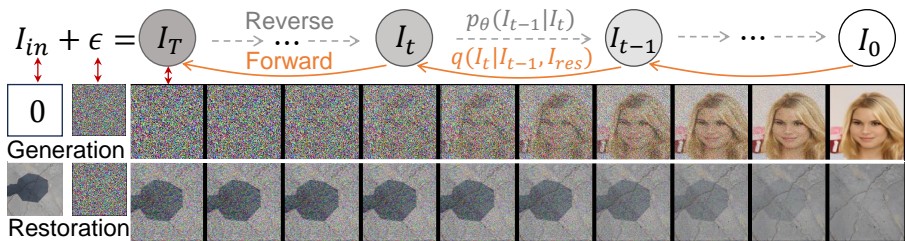

Figure 1: The proposed residual denoising diffusion model (RDDM) is a unified framework for image generation and restoration (a shadow removal task is shown here). We introduce residuals ($I_{res}$) in RDDM, redefining the forward diffusion process to involve simultaneous diffusion of residuals and noise. The residuals ($I_{res} = I_{in} - I_0$) diffusion represents the directional diffusion from the target image $I_0$ to the degraded input image $I_{in}$, while the noise ($\epsilon$) diffusion represents the random perturbations in the diffusion process. In RDDM, $I_0$ gradually diffuses into $I_T = I_{in} + \epsilon$, $\epsilon \sim \mathcal{N}(\mathbf{0}, \mathbf{I})$. In the third columns, $I_T$ is a purely noisy image for image generation since $I_{in} = 0$, and a noise-carrying degraded image for image restoration as $I_{in}$ is the degraded image.

schedule within a certain range during testing does not affect the image generation results, and removing the residuals firstly, followed by denoising, can also produce semantically consistent images. Our RDDM is compatible with widely used denoising diffusion models, i.e., our sampling process is consistent with that of DDPM (Ho et al., 2020) and DDIM (Song et al., 2021a) by transforming coefficient schedules. In addition, our RDDM natively supports conditional inputs, enabling networks trained with only an $\ell_1$ loss and a batch size of 1 to compete with state-of-the-art image restoration methods. We envision that our models can facilitate a unified and interpretable image-to-image distribution transformation methodology, highlighting that residuals and noise are equally important for diffusion models, e.g., the residual prioritizes certainty while the noise emphasizes diversity.

## 2   BACKGROUND

Denoising diffusion models (Ho et al., 2020; Sohl-Dickstein et al., 2015) aim to learn a distribution $p_\theta(I_0) := \int p_\theta(I_{0:T}) dI_{1:T}$[1] to approximate a target data distribution $q(I_0)$, where $I_0$ are target images and $I_1, \ldots, I_T$ ($T = 1000$) are latents of the same dimension as $I_0$. In the forward process, $q(I_0)$ is diffused into a Gaussian noise distribution using a fixed Markov chain,

$$q(I_{1:T}|I_0) := \prod_{t=1}^T q(I_t|I_{t-1}), \qquad q(I_t|I_{t-1}) := \mathcal{N}(I_t; \sqrt{\alpha_t}I_{t-1}, (1-\alpha_t)\mathbf{I}), \qquad (1)$$

where $\alpha_{1:T} \in (0, 1]^T$. $q(I_t|I_{t-1})$ can also be written as $I_t = \sqrt{\alpha_t}I_{t-1} + \sqrt{1 - \alpha_t}\epsilon_{t-1}$. In fact, it is simpler to sampling $I_t$ from $I_0$ by reparameterization (Kingma & Welling, 2014; 2019),

$$I_t = \sqrt{\bar{\alpha}_t}I_0 + \sqrt{1 - \bar{\alpha}_t}\epsilon, \text{ where } \epsilon \sim \mathcal{N}(\mathbf{0}, \mathbf{I}), \bar{\alpha}_t := \prod_{s=1}^t \alpha_s. \qquad (2)$$

The reverse process is also a Markov chain starting at $p_\theta(I_T) \sim \mathcal{N}(I_T; \mathbf{0}, \mathbf{I})$,

$$p_\theta(I_{0:T}) := p_\theta(I_T)\prod_{t=1}^T p_\theta(I_{t-1}|I_t), \qquad p_\theta(I_{t-1}|I_t) := \mathcal{N}(I_{t-1}; \mu_\theta(I_t, t), \Sigma_t \mathbf{I}), \qquad (3)$$

where $p_\theta(I_{t-1}|I_t)$ is a learnable transfer probability (the variance schedule $\Sigma_t$ is fixed). Ho et al. (2020) derive a simplified loss function from the maximum likelihood of $p_\theta(I_0)$, i.e., $L(\theta) := \mathbb{E}_{I_0 \sim q(I_0), \epsilon \sim \mathcal{N}(\mathbf{0}, \mathbf{I})}\left[\|\epsilon - \epsilon_\theta(I_t, t)\|^2\right]$. The estimated noise $\epsilon_\theta$ can be used to represent $\mu_\theta$ in $p_\theta(I_{t-1}|I_t)$, thus $I_{t-1}$ can be sampled from $p_\theta(I_{t-1}|I_t)$ step by step.

## 3   RESIDUAL DENOISING DIFFUSION MODELS

Our goal is to develop a dual diffusion process to unify and interpret image generation and restoration. We modify the representation of $I_T = \epsilon$ in traditional DDPM to $I_T = I_{in} + \epsilon$ in our RDDM, where $I_{in}$ is a degraded image (e.g., a shadow, low-light, or blurred image) for image restoration and is set to 0 for image generation. This modification is compatible with the widely used denoising diffusion model, e.g., $I_T = 0 + \epsilon$ is the pure noise ($\epsilon$) for generation. For image restoration, $I_T$ is a

---

[1]To understand diffusion from an image perspective, we use $I$ instead of $x$ in DDPM (Ho et al., 2020).

noisy-carrying degraded image $(I_{in} + \epsilon)$, as shown in the third column in Fig. 1. The modified forward process from $I_0$ to $I_T = I_{in} + \epsilon$ involves progressively degrading $I_0$ to $I_{in}$, and injecting noise $\epsilon$. This naturally results in a dual diffusion process, a residual diffusion to model the transition from $I_0$ to $I_{in}$ and a noise diffusion. For example, the forward diffusion process from the shadow-free image $I_0$ to the noisy carrying shadow image $I_T$ involves progressively adding shadows and noise, as shown in the second row in Fig. 1.

In the following subsections, we detail the underlying theory and the methodology behind our RDDM. Inspired by residual learning (He et al., 2016), we redefine each forward diffusion process step in Section 3.1. For the reverse process, we present a training objective to predict the residuals and noise injected in the forward process in Section 3.2. In Section 3.3, we propose three sampling methods, i.e., residual prediction (SM-Res), noise prediction (SM-N), and "residual and noise prediction" (SM-Res-N).

### 3.1 DIRECTIONAL RESIDUAL DIFFUSION PROCESS WITH PERTURBATION

To model the gradual degradation of image quality and the increment of noise, we define the single forward process step in our RDDM as follows:

$$I_t = I_{t-1} + I_{res}^t, \qquad I_{res}^t \sim \mathcal{N}(\alpha_t I_{res}, \beta_t^2 \mathbf{I}), \tag{4}$$

where $I_{res}^t$ represents a directional mean shift (residual diffusion) with random perturbation (noise diffusion) from state $I_{t-1}$ to state $I_t$, the residuals $I_{res}$ in $I_{res}^t$ is the difference between $I_{in}$ and $I_0$ (i.e., $I_{res} = I_{in} - I_0$), and two independent coefficient schedules $\alpha_t$ and $\beta_t$ control the residual and noise diffusion, respectively. In fact, it is simpler to sample $I_t$ from $I_0$ (like Eq. 2),

$$
\begin{aligned}
I_t =& I_{t-1} + \alpha_t I_{res} + \beta_t \epsilon_{t-1}, \text{where } \epsilon_{t-1}, \epsilon_{t-2} \ldots \epsilon \sim \mathcal{N}(\mathbf{0}, \mathbf{I}) \\
=& I_{t-2} + (\alpha_{t-1} + \alpha_t) I_{res} + (\sqrt{\beta_{t-1}^2 + \beta_t^2}) \epsilon_{t-2} \\
=& \ldots \\
=& I_0 + \bar{\alpha}_t I_{res} + \bar{\beta}_t \epsilon,
\end{aligned}
\tag{5}
$$

where $\bar{\alpha}_t = \sum_{i=1}^t \alpha_i$ and $\bar{\beta}_t = \sqrt{\sum_{i=1}^t \beta_i^2}$. If $t = T$, $\bar{\alpha}_T = 1$ and $I_T = I_{in} + \bar{\beta}_T \epsilon$. $\bar{\beta}_T$ can control the intensity of noise perturbation for image restoration (e.g., $\bar{\beta}_T^2 = 0.01$ for shadow removal), while $\bar{\beta}_T^2 = 1$ for image generation. The newly defined diffusion process via Eq. 5 has the sum-constrained variance, while DDPM has preserving variance (see Appendix A.4 and Fig. 7). From Eq. 4, the joint probability distributions in the forward process can be defined as:

$$q(I_{1:T}|I_0, I_{res}) := \prod_{t=1}^T q(I_t|I_{t-1}, I_{res}), \quad q(I_t|I_{t-1}, I_{res}) := \mathcal{N}(I_t; I_{t-1} + \alpha_t I_{res}, \beta_t^2 \mathbf{I}). \tag{6}$$

Eq. 5 defines the marginal probability distribution $q(I_t|I_0, I_{res}) = \mathcal{N}(I_t; I_0 + \bar{\alpha}_t I_{res}, \bar{\beta}_t^2 \mathbf{I})$. In fact, the forward diffusion of our RDDM is a mixture of three terms (i.e., $I_0$, $I_{res}$, and $\epsilon$), extending beyond the widely used denoising diffusion model that is a mixture of two terms, i.e, $I_0$ and $\epsilon$. A similar mixture form of three terms can be seen in several concurrent works, e.g., InDI (Delbracio & Milanfar, 2023), I2SB (Liu et al., 2023a), and IR-SDE (Luo et al., 2023).

### 3.2 GENERATION PROCESS AND TRAINING OBJECTIVE

In the forward process (Eq. 5), residuals ($I_{res}$) and noise ($\epsilon$) are gradually added to $I_0$, and then synthesized into $I_t$, while the reverse process from $I_T$ to $I_0$ involves the estimation of the residuals and noise injected in the forward process. We can train a residual network $I_{res}^\theta(I_t, t, I_{in})$ to predict $I_{res}$ and a noise network $\epsilon_\theta(I_t, t, I_{in})$ to estimate $\epsilon$. Using Eq. 5, we obtain the estimated target images $I_0^\theta = I_t - \bar{\alpha}_t I_{res}^\theta - \bar{\beta}_t \epsilon_\theta$. If $I_0^\theta$ and $I_{res}^\theta$ are given, the generation process is defined as,

$$p_\theta(I_{t-1}|I_t) := q_\sigma(I_{t-1}|I_t, I_0^\theta, I_{res}^\theta), \tag{7}$$

where the transfer probability $q_\sigma(I_{t-1}|I_t, I_0, I_{res})^2$ from $I_t$ to $I_{t-1}$ is,

$$q_\sigma(I_{t-1}|I_t, I_0, I_{res}) = \mathcal{N}\left(I_{t-1}; I_0 + \bar{\alpha}_{t-1} I_{res} + \sqrt{\bar{\beta}_{t-1}^2 - \sigma_t^2} \frac{I_t - (I_0 + \bar{\alpha}_t I_{res})}{\bar{\beta}_t}, \sigma_t^2 \mathbf{I}\right), \tag{8}$$

---

[2]Eq. 8 does not change $q(I_t|I_0, I_{res})$ in Appendix A.2.

where $\sigma_t^2 = \eta \beta_t^2 \bar{\beta}_{t-1}^2 / \bar{\beta}_t^2$ and $\eta$ controls whether the generation process is random ($\eta = 1$) or deterministic ($\eta = 0$). Using Eq. 7 and Eq. 8, $I_{t-1}$ can be sampled from $I_t$ via:

$$I_{t-1} = I_t - (\bar{\alpha}_t - \bar{\alpha}_{t-1})I_{res}^\theta - (\bar{\beta}_t - \sqrt{\bar{\beta}_{t-1}^2 - \sigma_t^2})\epsilon_\theta + \sigma_t\epsilon_t, \text{where } \epsilon_t \sim \mathcal{N}(\mathbf{0}, \mathbf{I}). \quad (9)$$

When $\eta = 0$ (i.e., $\sigma_t = 0$), the sampling process is deterministic,

$$I_{t-1} = I_t - (\bar{\alpha}_t - \bar{\alpha}_{t-1})I_{res}^\theta - (\bar{\beta}_t - \bar{\beta}_{t-1})\epsilon_\theta. \quad (10)$$

We derive the following simplified loss function for training (Appendix A.1):

$$L_{res}(\theta) := \mathbb{E}\left[\lambda_{res} \left\| I_{res} - I_{res}^\theta(I_t, t, I_{in}) \right\|^2\right], \qquad L_\epsilon(\theta) := \mathbb{E}\left[\lambda_\epsilon \left\| \epsilon - \epsilon_\theta(I_t, t, I_{in}) \right\|^2\right], \quad (11)$$

where the hyperparameters $\lambda_{res}, \lambda_\epsilon \in \{0, 1\}$, and the training input image $I_t$ is synthesized using $I_0$, $I_{res}$, and $\epsilon$ by Eq. 5. $I_t$ can also be synthesized using $I_{in}$ (replace $I_0$ in Eq. 5 by $I_0 = I_{in} - I_{res}$),

$$I_t = I_{in} + (\bar{\alpha}_t - 1)I_{res} + \bar{\beta}_t\epsilon. \quad (12)$$

## 3.3 Sampling Method Selection Strategies

For the generation process (from $I_t$ to $I_{t-1}$), $I_t$ and $I_{in}$ are known, and thus $I_{res}$ and $\epsilon$ can represent each other by Eq. 12. From Eq. 11 and Eq. 12, we propose three sampling methods as follows.
**SM-Res.** When $\lambda_{res} = 1$ and $\lambda_\epsilon = 0$, the residuals $I_{res}^\theta$ are predicted by a network, while the noise $\epsilon_\theta$ is represented as a transformation of $I_{res}^\theta$ using Eq. 12.
**SM-N.** When $\lambda_{res} = 0$ and $\lambda_\epsilon = 1$, the noise $\epsilon_\theta$ is predicted by a network, while the residuals $I_{res}^\theta$ are represented as a transformation of $\epsilon_\theta$ using Eq. 12.
**SM-Res-N.** When $\lambda_{res} = 1$ and $\lambda_\epsilon = 1$, both the residuals and the noise are predicted.
To determine the optimal sampling method for real-world applications, we give empirical strategies and automatic selection algorithms in the following.

Table 1: Sampling method analysis. The sampling steps are 10 on the CelebA $64 \times 64$ (Liu et al., 2015) dataset, 5 on the ISTD (Wang et al., 2018) dataset, 2 on the LOL (Wei et al., 2018) dataset, and 5 on the RainDrop (Qian et al., 2018) dataset.

| Sampling Method | Generation (CelebA) | | Shadow removal (ISTD) | | | Low-light (LOL) | | Deraining (RainDrop) | |
|---|---|---|---|---|---|---|---|---|---|
| | FID ($\downarrow$) | IS ($\uparrow$) | MAE($\downarrow$) | PSNR($\uparrow$) | SSIM($\uparrow$) | PSNR($\uparrow$) | SSIM($\uparrow$) | PSNR($\uparrow$) | SSIM($\uparrow$) |
| SM-Res | 31.47 | 1.73 | 4.76 | 30.72 | 0.959 | **25.39** | **0.937** | 19.15 | 0.7179 |
| SM-N | **23.25** | **2.05** | 81.01 | 11.34 | 0.175 | 16.30 | 0.649 | 31.96 | 0.9509 |
| SM-Res-N | 28.90 | 1.78 | **4.67** | **30.91** | **0.962** | 23.90 | 0.931 | **32.51** | **0.9563** |

**Empirical Research.** Table 1 presents that the SM-Res shows better results for image restoration but offers a poorer FID for generation. On the other hand, the SM-N yields better frechet inception distance (FID in Heusel et al. (2017)) and inception scores (IS), but is ineffective in image restoration (e.g., PSNR 11.34 for shadow and 16.30 for low-light). This may be due to the inadequacy of using $\epsilon_\theta$ to represent $I_{res}^\theta$ in Eq. 12 for restoration tasks. We attribute these inconsistent results to the fact that **residual predictions prioritize certainty, whereas noise predictions emphasize diversity**. In our experiments, we use SM-Res for low-ligh enhancement, SM-N for image generation, and SM-Res-N for other image restoration tasks. For an unknown new task, we empirically recommend using SM-Res for tasks that demand higher certainty and SM-N for those requiring greater diversity.

**Automatic Target Domain Prediction Strategies (ATDP).** To automatically choose between SM-Res or SM-N for an unknown task, we develop an automatic sampling selection algorithm in Appendix B.2. This algorithm requires only a single network and learns the hyperparameter in Eq. 11, enabling a gradual transition from combined residual and noise training (akin to SM-Res-N) to individual prediction (SM-Res or SM-N). This plug-and-play training strategy requires less than 1000 additional training iterations and is fully compatible with the current denoising-based diffusion methods (Ho et al., 2020). Our RDDM using ATDP has the potential to provide a unified and interpretable methodology for modeling, training, and inference pipelines for unknown target tasks.

**Comparison with Other Prediction Methods.** Our SM-N is similar to DDIM (Song et al., 2021a) (or DDPM (Ho et al., 2020)), which only estimates the noise, and is consistent with DDPM and DDIM by transforming the coefficient schedules in Eq. 9 (the proof in Appendix A.3),

$$\bar{\alpha}_t = 1 - \sqrt{\bar{\alpha}_{DDIM}^t}^3, \qquad \bar{\beta}_t = \sqrt{1 - \bar{\alpha}_{DDIM}^t}, \qquad \sigma_t^2 = \sigma_t^2(DDIM). \quad (13)$$

---

[3]$\bar{\alpha}_{DDIM}^t$ here is $\alpha_t$ of DDIM (Song et al., 2021a).

In fact, current research has delved into numerous diffusion forms that extend beyond noise estimation. For example, IDDPM (Nichol & Dhariwal, 2021) proposes that it is feasible to estimate noise ($\epsilon_\theta$), clean target images ($I_0^\theta$), or the mean term ($\mu_\theta$) to represent the transfer probabilities (i.e., $p_\theta(I_{t-1}|I_t)$ in Eq. 3). The score-based generative model (SGM) (Song & Ermon, 2019) and Schrödinger Bridge (I2SB (Liu et al., 2023a)) estimate the score of noisy data (i.e., the sum of residuals and noise $\sum_{i=1}^t I_{res}^t$). ColdDiffusion (Bansal et al., 2022), InDI (Delbracio & Milanfar, 2023), and consistency models (Song et al., 2023) estimate the clean target images ($I_0$). Rectified Flow (Liu et al., 2023d) predicts the residuals ($I_{res}$) to align with the image linear interpolation process without noise diffusion (i.e., $I_T = I_{in}$). A detailed comparison can be found in Appendix A.5.

These previous/concurrent works choose to estimate the noise, the residual, the target image, or its linear transformation term. In contrast, we introduce residual estimation while also embracing noise for both generation and restoration. Residuals and noise have equal and independent status, which is reflected in the forward process (Eq. 5), the reverse process (Eq. 10), and the loss function (Eq. 11). This independence means that the noise diffusion can even be removed and only the residual diffusion retained to model the image interpolation process (when $\bar{\beta}_T = 0$ in Eq. 5, RDDM degenerates to Rectified Flow (Liu et al., 2023d)). In addition, this property derives an independent dual diffusion framework in Section 4.

## 4 DECOUPLED DUAL DIFFUSION FRAMEWORK

Upon examining DDPM from the perspective of RDDM, we discover that DDPM indeed involves the simultaneous diffusion of residuals and noise, which is evident as Eq. 43 becomes equivalent to Eq. 39 in Appendix A.3. We find that it is possible to decouple these two types of diffusion. Section 4.1 presents a decoupled forward diffusion process. In Section 4.2, we propose a partially path-independent generation process and decouple the simultaneous sampling into first removing the residuals and then removing noise (see Fig. 3(d) and Fig. 16). This decoupled dual diffusion framework sheds light on the roles of deresidual and denoising in the DDPM generation process.

### 4.1 DECOUPLED FORWARD DIFFUSION PROCESS

Our defined coefficients ($\alpha_t, \beta_t^2$) offer a distinct physical interpretation. In the forward diffusion process (Eq. 5), $\alpha_t$ controls the speed of residual diffusion and $\beta_t^2$ regulates the speed of noise diffusion. In the reverse generation process (Eq. 10), $\bar{\alpha}_t$ and $\bar{\beta}_t$ are associated with the speed of removing residual and noise, respectively. In fact, there are no constraints on $\alpha^t$ and $\beta_t^2$ in Eq. 5, meaning that the residual diffusion and noise diffusion are independent of each other. Utilizing this decoupled property and the difference between these two diffusion processes, we should be able to design a better coefficient schedule, e.g., $\alpha_t$ (linearly decreasing) and $\beta_t^2$ (linearly increasing) in Table 2. This aligns with the intuition that, during the reverse generation process (from $T$ to 0), the estimated residuals become increasingly accurate while the estimated noise should also

Table 2: Coefficient schedules analysis on CelebA ($64 \times 64$) (Liu et al., 2015). In our RDDM, the residual diffusion and noise diffusion are decoupled, so one may design a better schedule in the decoupled coefficient space, e.g., $\alpha_t$ (linearly decreasing), $\beta_t^2$ (linearly increasing). To be fair, all coefficient schedules were retrained using the same network structure, training, and evaluation. The sampling method is SM-N with 10 sampling steps using Eq. 10.

| Schedules | FID ($\downarrow$) | IS ($\uparrow$) |
|---|---|---|
| Linear (DDIM Song et al. (2021a)) | 28.39[4] | 2.05 |
| Scaled linear (Rombach et al., 2022) | 28.15 | 2.00 |
| Squared cosine (Nichol & Dhariwal, 2021) | 47.21 | **2.64** |
| $\alpha_t$ (mean), $\beta_t^2$ (mean) | 38.35 | 2.22 |
| $\alpha_t$ (linearly increasing), $\beta_t^2$ (linearly increasing) | 40.03 | 2.45 |
| $\alpha_t$ (linearly decreasing), $\beta_t^2$ (linearly decreasing) | 27.82 | 2.26 |
| $\alpha_t$ (linearly decreasing), $\beta_t^2$ (linearly increasing) | **23.25** | 2.05 |

weaken progressively. Therefore, when $t$ is close to 0, the deresidual pace should be faster and the denoising pace should be slower. Since our $\alpha_t$ and $\beta_t^2$ represent the speed of diffusion, we name the curve in Fig. 6 (b-d) (see Appendix A.3) the *diffusion speed curve*.

---

[4]Our RDDM is implemented based on the popular diffusion repository github.com/lucidrains/denoising-diffusion-pytorch. Differences in network structure and training details may lead to poorer FID. We have verified sampling consistency with DDIM (Song et al., 2021a) in Table 3(a) and Appendix A.3.

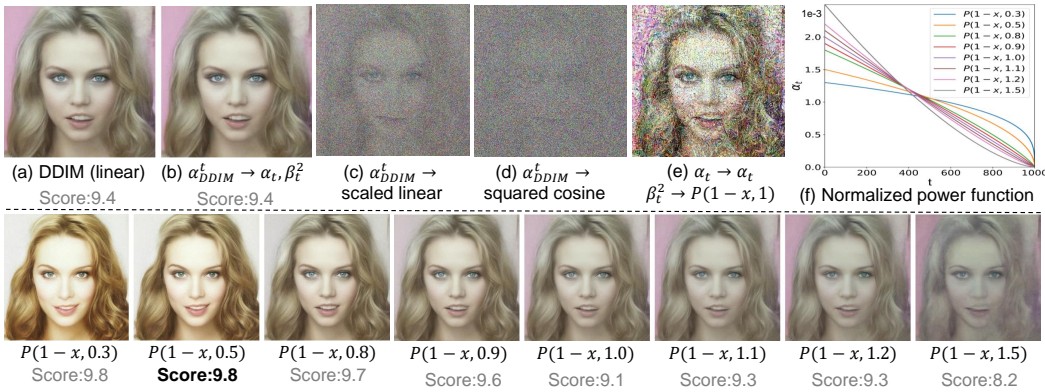

(a) DDIM (linear)
Score:9.4

(b) $\alpha_{DDIM}^t \to \alpha_t, \beta_t^2$
Score:9.4

(c) $\alpha_{DDIM}^t \to$ scaled linear

(d) $\alpha_{DDIM}^t \to$ squared cosine

(e) $\alpha_t \to \alpha_t$ $\beta_t^2 \to P(1-x, 1)$

(f) Normalized power function

$P(1-x, 0.3)$
Score:9.8

$P(1-x, 0.5)$
**Score:9.8**

$P(1-x, 0.8)$
Score:9.7

$P(1-x, 0.9)$
Score:9.6

$P(1-x, 1.0)$
Score:9.1

$P(1-x, 1.1)$
Score:9.3

$P(1-x, 1.2)$
Score:9.3

$P(1-x, 1.5)$
Score:8.2

(g) First convert $\alpha_{DDIM}^t$ to $\alpha_t, \beta_t^2$ and then readjust the converted $\alpha_t$ without touching the $\beta_t^2$

Figure 2: Analysis of readjusting coefficient schedules. We find that changing the $\alpha_t$ schedule barely affects the denoising process in (g) and edited faces may have higher face scores when assessed using AI face scoring software[5]. These images were generated using a pre-trained UNet on the CelebA ($256 \times 256$) dataset (Liu et al., 2015) with 10 sampling steps.

## 4.2 PARTIALLY PATH-INDEPENDENT GENERATION PROCESS

In the original DDPM (Ho et al., 2020) or DDIM (Song et al., 2021a), when the $\alpha_{DDIM}^t$ schedule changes, it is necessary to retrain the denoising network because this alters the diffusion process (Rombach et al., 2022; Nichol & Dhariwal, 2021). As shown in Fig. 2(c)(d), directly changing the $\alpha_{DDIM}^t$ schedule causes denoising to fail. Here, we propose a path-independent generation process, i.e., modifying the diffusion speed curve does not cause the image generation process to fail. We try to readjust the diffusion speed curve in the generation process. First, we convert the $\alpha_{DDIM}^t$ schedule of a pre-trained DDIM into the $\alpha_t$ and $\beta_t^2$ schedules of our RDDM using Eq. 13 (from Fig. 2(a) to Fig. 2(b). We then readjust the converted $\alpha_t$ schedules using the normalized power function ($P(x, a)$ in Fig. 2(f)), without touching the $\beta_t^2$ schedule that controls noise diffusion, as shown in Fig. 2(g). $P(x, a)$ is defined as ($a$ is a parameter of the power function),

$$P(x, a) := x^a / \int_0^1 x^a dx, \text{where } x = t/T. \tag{14}$$

These schedule modifications shown in Fig. 2 lead to the following key findings.

**1.** Fig. 2(g) shows that modifying the residual diffusion speed curve ($\alpha_t$) leads to a drastic change in the generation results, probably due to $I_{res}^\theta$ being represented as a transformation of $\epsilon_\theta$ using Eq. 12.

**2.** As the time condition $t$ represents the current noise intensity in the denoising network ($\epsilon_\theta(I_t, t, 0)$), modifying the noise diffusion speed curve ($\beta_t^2$) causes $t$ to deviate from accurately indicating the current noise intensity, leading to denoising failure, as shown in Fig. 2(e).

Nonetheless, we believe that, corresponding to the decoupled forward diffusion process, there should also be a path-independent reverse generation process. To develop a path-independent generation process, we improve the generation process based on the above two key findings:

**1.** Two networks are used to estimate $I_{res}^\theta$ and $\epsilon_\theta$ separately, i.e., SM-Res-N-2Net in Appendix B.2.

**2.** $\bar{\alpha}_t$ and $\bar{\beta}_t$ are used for the time conditions embedded in the network, i.e., $I_{res}^\theta(I_t, t, 0) \to I_{res}^\theta(I_t, \bar{\alpha}_t \cdot T, 0)$, $\epsilon_\theta(I_t, t, 0) \to \epsilon_\theta(I_t, \bar{\beta}_t \cdot T, 0)$.

These improvements lead to a partially path-independent generation process, as evidenced by the results shown in Fig. 3(c).

**Analysis of Partially Path-independence via Green's Theorem.** "Path-independence" reminds us of Green's theorem in curve integration (Riley et al., 2006). From Eq. 10, we have:

$$I_t - I_{t-1} = (\bar{\alpha}_t - \bar{\alpha}_{t-1})I_{res}^\theta + (\bar{\beta}_t - \bar{\beta}_{t-1})\epsilon_\theta,$$
$$\mathrm{d}I(t) = I_{res}^\theta(I(t), \bar{\alpha}(t) \cdot T, 0)\mathrm{d}\bar{\alpha}(t) + \epsilon_\theta(I(t), \bar{\beta}(t) \cdot T, 0)\mathrm{d}\bar{\beta}(t), \tag{15}$$

---

[5]https://ux.xiaoice.com/beautyv3

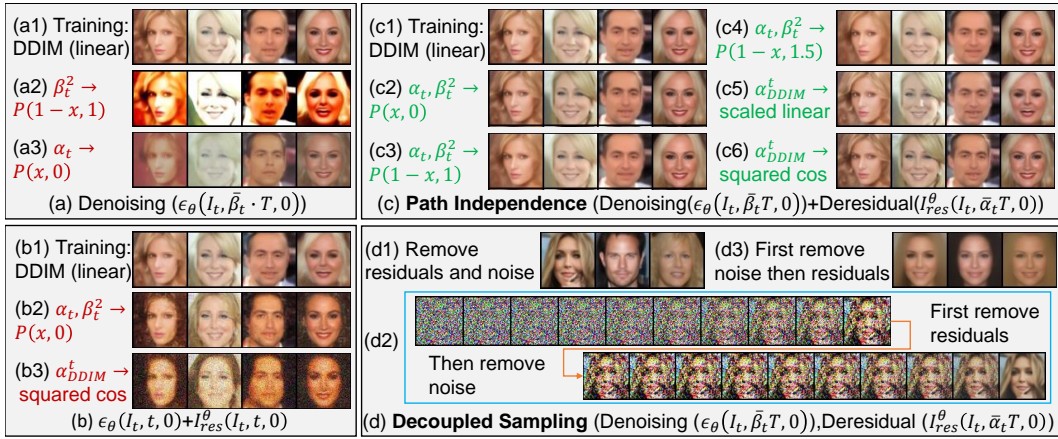

Figure 3: Partially path-independent generation process. (a1) We trained a denoising network using the DDIM linear schedule (Song et al., 2021a). (a2-a3) We modified the $\alpha_t$ and $\beta_t^2$ schedules during testing. (b) We trained two networks to remove noise and residuals. In contrast to the sharply varying images in (a2-a3) and the noisy images in (b2-b3), (c) shows that we constructed a path independent generation process where modifications to the diffusion speed curve can generate a noise-free image with little variation in image semantics. (d) The simultaneous sampling in (d1) or (c) can be decomposed into first removing residuals and then noise (d2), or removing noise and then residuals (d3). In (d3), diversity is significantly reduced because noise is removed first.

where $I(t) = I(0) + \bar{\alpha}(t)I_{res} + \bar{\beta}(t)\epsilon$. Given inputs $I(t)$ and $\bar{\alpha}(t)$, the denoising network learns to approximate the noise $\epsilon$ in $I(t)$ by estimating $\epsilon_\theta$. If this network is trained well and robust enough, it should be able to avoid the interference of the residual terms $\bar{\alpha}(t)I_{res}$ in $I(t)$. This also applies to a robust residual estimation network. Thus, we have

$$\frac{\partial I_{res}^\theta(I(t), \bar{\alpha}(t) \cdot T)}{\partial \bar{\beta}(t)} \approx 0, \qquad \frac{\partial \epsilon_\theta(I(t), \bar{\beta}(t) \cdot T)}{\partial \bar{\alpha}(t)} \approx 0. \tag{16}$$

If the equation in Formula 16 holds true, it serves as a necessary and sufficient condition for path independence in curve integration, which provides an explanation for why Fig. 3(c) achieves a partially path-independent generation process. The path-independent property is related to the network's resilience to disturbances and applies to disturbances that vary within a certain range. However, excessive disturbances can lead to visual inconsistencies, e.g., readjusting $\alpha_t$ and $\beta_t^2$ to $P(x, 5)$. Thus, we refer to this generative property as partially path-independent. We also investigated two reverse paths to gain insight into the implications of the proposed partial path independence. In the first case, the residuals are removed first, followed by the noise: $I(T) \xrightarrow{-I_{res}} I(0) + \bar{\beta}_T\epsilon \xrightarrow{-\bar{\beta}_T\epsilon} I(0)$. The second case involves removing the noise first and then the residuals: $I(T) \xrightarrow{-\bar{\beta}_T\epsilon} I_{in} \xrightarrow{-I_{res}} I(0)$. The first case (Fig. 3(d2)) shows that removing residuals controls semantic transitions, while the second case (Fig. 3(d3)) shows that diversity is significantly reduced because noise is removed first. Fig. 3(d) validates our argument that residuals control directional semantic drift (certainty) and noise controls random perturbation (diversity). See Appendix B.4 for more details.

## 5 EXPERIMENTS

**Image Generation.** We can convert a pre-trained[6] DDIM (Song et al., 2021a) to RDDM by coefficient transformation using Eq. 13, and generate images by Eq. 9. Table 3(a) verifies that the quality of the generated images before and after the conversion is nearly the same[7]. We show the generated face images with 10 sampling steps in Fig. 4(a).

**Image Restoration.** We extensively evaluate our method on several image restoration tasks, including shadow removal, low-light enhancement, deraining, and deblurring on 5 datasets. Notably,

---

[6]https://huggingface.co/google/ddpm-celebahq-256

[7]The subtle differences in larger sampling steps may stem from errors introduced by numerical representation limitations during coefficient transformation, which may accumulate and amplify in larger sampling steps.

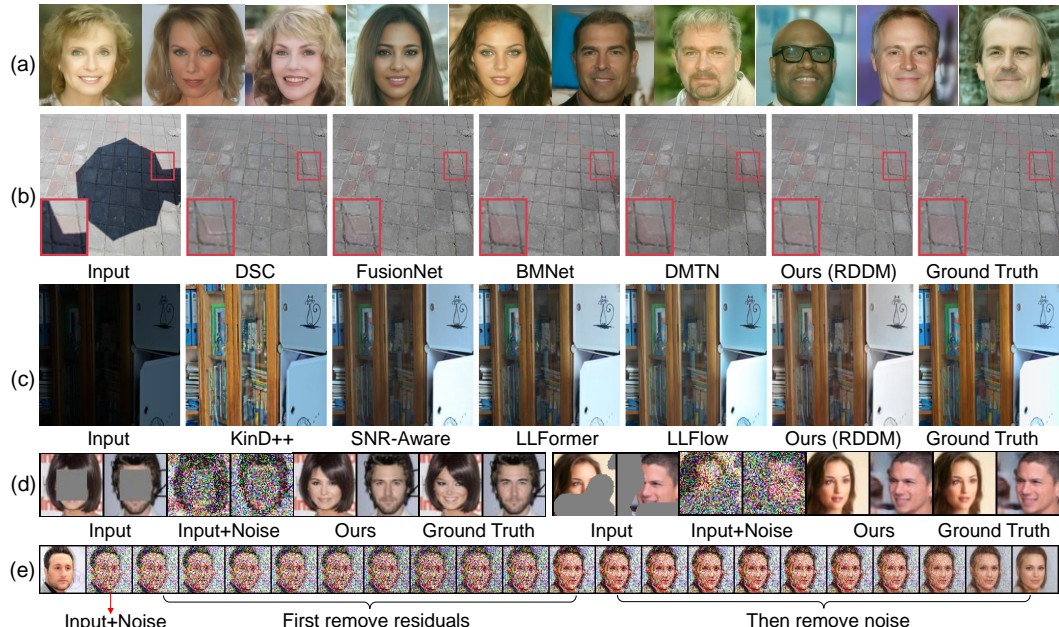

Figure 4: Application of our RDDM. (a) Image generation on the CelebA dataset (Liu et al., 2015). (b) Shadow removal on the ISTD dataset (Wang et al., 2018). (c) Low-light enhancement on the LOL dataset (Wei et al., 2018). (d) Image inpainting (center and irregular mask). (e) The image translation process can be regarded as first translating the semantics and then generating the details.

Table 3: Quantitative comparison results of image generation on the CelebA ($256 \times 256$) dataset (Liu et al., 2015), shadow removal on the ISTD dataset (Wang et al., 2018), low-light enhancement on the LOL (Wei et al., 2018) dataset, and deraining on the RainDrop (Qian et al., 2018) dataset. "S, NS, ALL" in (b) denote shadow area (S), non-shadow area (NS) and whole image (ALL).

| (a) CelebA (FID) | DDIM | DDIM→RDDM | (b) Shadow Removal | MAE(↓) | | | SSIM(↑) | | | PSNR(↑) | | |
|---|---|---|---|---|---|---|---|---|---|---|---|---|
| | | | | S | NS | ALL | S | NS | ALL | S | NS | ALL |
| 5 steps | 69.60 | 69.60 | DSC (Hu et al., 2020) ¶ | 9.48 | 6.14 | 6.67 | 0.967 | - | - | 33.45 | - | - |
| 10 steps | 40.45 | 40.41 | FusionNet (Fu et al., 2021) | 7.77 | 5.56 | 5.92 | 0.975 | 0.880 | 0.945 | 34.71 | 28.61 | 27.19 |
| 15 steps | 32.67 | 32.71 | BMNet (Zhu et al., 2022a) | 7.60 | 4.59 | 5.02 | 0.988 | 0.976 | 0.959 | 35.61 | 32.80 | 30.28 |
| 20 steps | 30.61 | 30.77 | DMTN (Liu et al., 2023b) | 7.00 | 4.28 | 4.72 | 0.990 | 0.979 | 0.965 | 35.83 | 33.01 | 30.42 |
| 100 steps | 23.66 | 24.92 | Ours (RDDM) | 6.67 | 4.27 | 4.67 | 0.988 | 0.979 | 0.962 | 36.74 | 33.18 | 30.91 |

| (c) Low-light | PSNR(↑) | SSIM(↑) | LPIPS (↓) | (d) Deraining | PSNR(↑) | SSIM(↑) |
|---|---|---|---|---|---|---|
| KinD++ (Zhang et al., 2021) | 17.752 | 0.760 | 0.198 | AttnGAN (Qian et al., 2018) | 31.59 | 0.9170 |
| KinD++-SKF (Yuhui et al., 2023) | 20.363 | 0.805 | 0.201 | DuRN (Liu et al., 2019) | 31.24 | 0.9259 |
| DCC-Net (Zhang et al., 2022) | 22.72 | 0.81 | - | RainAttn (Quan et al., 2019) | 31.44 | 0.9263 |
| SNR-Aware (Xu et al., 2022) | 24.608 | 0.840 | 0.151 | IDT (Xiao et al., 2022) | 31.87 | 0.9313 |
| LLFlow (Wang et al., 2022a) | 25.19 | 0.93 | 0.11 | RainDiff64 (Özdenizci & Legenstein, 2023) | 32.29 | 0.9422 |
| LLFormer (Wang et al., 2023) | 23.649 | 0.816 | 0.169 | RainDiff128 (Özdenizci & Legenstein, 2023) | 32.43 | 0.9334 |
| Ours (RDDM) | 25.392 | 0.937 | 0.134 | Ours (RDDM) | 32.51 | 0.9563 |

our RDDM uses an identical UNet and is trained with a batch size of 1 for all these tasks. In contrast, SOAT methods often involve elaborate network architectures, such as multi-stage (Fu et al., 2021; Zhu et al., 2022b; Wang et al., 2022a), multi-branch (Cun et al., 2020), Transformer (Wang et al., 2023), and GAN (Kupyn et al., 2019), or sophisticated loss functions like the chromaticity (Jin et al., 2021), texture similarity (Zhang et al., 2019), and edge loss (Zamir et al., 2021). Table 3 and Fig. 4(b-c) show that our RDDM is competitive with the SOTA restoration methods. See Appendix B for more training details and comparison results.

We extend DDPM (Ho et al., 2020)/DDIM (Song et al., 2021a), initially uninterpretable for image restoration, into a unified and interpretable diffusion model for both image generation and restoration by introducing residuals. However, the residual diffusion process represents the directional diffusion from target images to conditional input images, which does not involve a priori information about the image restoration task, and therefore is not limited to it. Beyond image generation and restoration, we show examples of image inpainting and image translation to verify that our

RDDM has the potential to be a unified and interpretable methodology for image-to-image distribution transformation. **We do not intend to achieve optimal performance on all tasks by tuning all hyperparameters.** The current experimental results show that RDDM 1) achieves consistent **image generation** performance with DDIM after coefficient transformation, 2) competes with state-of-the-art **image restoration** methods using a generic UNet with only an $\ell_1$ loss, a batch size of 1, and fewer than 5 sampling steps, and 3) has satisfactory visual results of **image inpainting** and **image translation** (see Fig. 4(d-e), Fig. 13, or Fig. 14), which have successfully validated the effectiveness of our RDDM.

## 6 RELATED WORK

Denoising diffusion models (e.g., DDPM (Ho et al., 2020), SGM (Song & Ermon, 2019; Song et al., 2021b), and DDIM (Song et al., 2021a)) were initially developed for image generation. Subsequent image restoration methods (Lugmayr et al., 2022; Rombach et al., 2022; Guo et al., 2023) based on DDPM and DDIM feed a degraded image as a conditional input to a denoising network, e.g., DvSR (Whang et al., 2022), SR3 (Saharia et al., 2022), and WeatherDiffusion (Özdenizci & Legenstein, 2023), which typically require large sampling steps and batch sizes. Additionally, the reverse process starting from noise in these methods seems unnecessary and inefficient for image restoration tasks. Thus, SDEdit (Meng et al., 2021a), ColdDiffusion (Bansal et al., 2022), InDI (Delbracio & Milanfar, 2023), and I2SB (Liu et al., 2023a) propose generating a clear image directly from a degraded image or noise-carrying degraded image. InDI (Delbracio & Milanfar, 2023) and I2SB (Liu et al., 2023a), which also present unified image generation and restoration frameworks, are the most closely related to our proposed RDDM. Specifically, the forward diffusion of InDI, I2SB, and our RDDM consistently employs a mixture of three terms (i.e., input images $I_{in}$, target images $I_0$, and noise $\epsilon$), extending beyond the denoising-based diffusion model (Ho et al., 2020; Song et al., 2021a) which incorporates a mixture of two terms (i.e., $I_0$ and $\epsilon$). However, InDI (Delbracio & Milanfar, 2023) and I2SB (Liu et al., 2023a) opt for estimating the target image or its linear transformation term to replace the noise estimation, akin to a special case of our RDDM (SM-Res). In contrast, we introduce residual estimation while also embracing noise for both generation and restoration tasks. Our RDDM can further extend DDPM (Ho et al., 2020), DDIM (Song et al., 2021a), InDI (Delbracio & Milanfar, 2023), and I2SB (Liu et al., 2023a) to independent double diffusion processes, and pave the way for the multi-dimensional diffusion process. We highlight that residuals and noise are equally important, e.g., the residual prioritizes certainty while the noise emphasizes diversity. In addition, our work is related to coefficient schedule design (Rombach et al., 2022; Nichol & Dhariwal, 2021), variance strategy optimization (Kingma et al., 2021; Nichol & Dhariwal, 2021; Bao et al., 2022b;a), superimposed image decomposition (Zou et al., 2020; Duan et al., 2022), curve integration (Riley et al., 2006), stochastic differential equations (Song et al., 2021b), and residual learning (He et al., 2016) for image restoration (Zhang et al., 2017; 2020; Anwar & Barnes, 2020; Zamir et al., 2021; Tu et al., 2022; Liu et al., 2023c). See Appendix A.5 for detailed comparison.

## 7 CONCLUSIONS AND DISCUSSIONS

We present a unified dual diffusion model called Residual Denoising Diffusion Models (RDDM) for image restoration and image generation. This is a three-term mixture framework beyond the previous denoising diffusion framework with two-term mixture. We demonstrate that our sampling process is consistent with that of DDPM and DDIM through coefficient schedule transformation, and propose a partially path-independent generation process. Our experimental results on four different image restoration tasks show that RDDM achieves SOTA performance in no more than five sampling steps. We believe that our model and framework hold the potential to provide a unified methodology for image-to-image distribution transformation and pave the way for the multi-dimensional diffusion process. However, there are certain limitations and areas for further investigation that should be addressed: (a) a deeper understanding of the relationship between our RDDM and curve/multivariate integration, (b) a diffusion model trained with one set of pre-trained parameters to handle several different tasks, (c) implementing adaptive learning coefficient schedules to reduce the sampling steps while improving the quality of the generated images, and (d) constructing multi-dimensional latent diffusion models for multimodal fusion and exploring interpretable text-to-image frameworks.

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

## APPENDIX

## A DERIVATIONS AND PROOFS

### A.1 PERTURBED GENERATION PROCESS

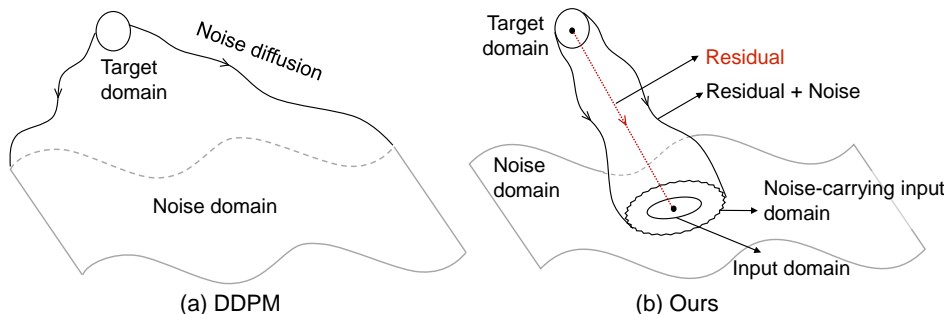

Figure 5: Denoising diffusion process - DDPM (Ho et al., 2020) (a) and our residual denoising diffusion process (b). For image restoration, we introduce residual diffusion to represent the diffusion direction from the target image to the input image.

Fig. 5 shows the difference between the forward diffusion process of DDPM (Ho et al., 2020) and our RDDM. Unlike the noise diffusion of DDPM (Ho et al., 2020), ours is a directional residual diffusion process with perturbation. Next, we derive the reverse sampling formula.

For the reverse generation process from $I_t$ to $I_{t-1}$, we can represent the transfer probabilities $q(I_{t-1}|I_t, I_0, I_{res})$ by Bayes' rule:

$$q(I_{t-1}|I_t, I_0, I_{res}) = q(I_t|I_{t-1}, I_0, I_{res}) \frac{q(I_{t-1}|I_0, I_{res})}{q(I_t|I_0, I_{res})}, \tag{17}$$

where $q(I_{t-1}|I_0, I_{res}) = \mathcal{N}(I_{t-1}; I_0 + \bar{\alpha}_{t-1}I_{res}, \bar{\beta}_{t-1}^2 \mathbf{I})$ from Eq. 5, and $q(I_t|I_{t-1}, I_0, I_{res}) = q(I_t|I_{t-1}, I_{res})^8 = \mathcal{N}(I_t; I_{t-1} + \alpha_t I_{res}, \beta_t^2 \mathbf{I})$ from Eq. 6. Thus, we have (considering only the exponential term)

$$q(I_{t-1}|I_t, I_0, I_{res}) = \mathcal{N}(I_{t-1}; \mu_t(x_t, I_0, I_{res}), \Sigma_t(x_t, I_0, I_{res})\mathbf{I})$$
$$\propto exp\left(-\frac{1}{2}\left(\frac{(I_t - I_{t-1} - \alpha_t I_{res})^2}{\beta_t^2} + \frac{(I_{t-1} - I_0 - \bar{\alpha}_{t-1}I_{res})^2}{\bar{\beta}_{t-1}^2}\right) - \frac{(I_t - I_0 - \bar{\alpha}_t I_{res})^2}{\bar{\beta}_t^2}\right)$$
$$= exp\left(-\frac{1}{2}\left(\left(\frac{\bar{\beta}_t^2}{\beta_t^2 \bar{\beta}_{t-1}^2}\right)I_{t-1}^2 - 2\left(\frac{I_t - \alpha_t I_{res}}{\beta_t^2} + \frac{\bar{\alpha}_{t-1}I_{res} + I_0}{\bar{\beta}_{t-1}^2}\right)I_{t-1} + C(I_t, I_0, I_{res})\right)\right), \tag{18}$$

where the $C(I_t, I_0, I_{res})$ term is not related to $I_{t-1}$. From Eq. 18, $\mu_t(x_t, I_0, I_{res})$ and $\Sigma_t(x_t, I_0, I_{res})$ are represented as follows,

$$\mu_t(x_t, I_0, I_{res}) = \left(\frac{I_t - \alpha_t I_{res}}{\beta_t^2} + \frac{\bar{\alpha}_{t-1}I_{res} + I_0}{\bar{\beta}_{t-1}^2}\right)/\frac{\bar{\beta}_t^2}{\beta_t^2 \bar{\beta}_{t-1}^2} \tag{19}$$

$$= \frac{\bar{\beta}_{t-1}^2}{\bar{\beta}_t^2}I_t + \frac{\beta_t^2 \bar{\alpha}_{t-1} - \bar{\beta}_{t-1}^2 \alpha_t}{\bar{\beta}_t^2}I_{res} + \frac{\beta_t^2}{\bar{\beta}_t^2}I_0 \tag{20}$$

$$= I_t - \alpha_t I_{res} - \frac{\beta_t^2}{\bar{\beta}_t}\epsilon, \tag{21}$$

$$\Sigma_t(x_t, I_0, I_{res}) = \frac{\beta_t^2 \bar{\beta}_{t-1}^2}{\bar{\beta}_t^2}. \tag{22}$$

---

[8]Each step in Eq. 5 adds a new random Gaussian noise in the random forward diffusion. Thus for simplicity, we assume $q(I_t|I_{t-1}, I_0, I_{res}) = q(I_t|I_{t-1}, I_{res})$, it follows that $I_0$ is not important for $I_t$ when $I_{t-1}$ presents as a condition.

Eq. 5 is used for the derivation from Eq. 20 to Eq. 21. Then, we define the generation process to start from $p_\theta(I_T) \sim \mathcal{N}(I_T; \mathbf{0}, \mathbf{I})$,

$$p_\theta(I_{t-1}|I_t) = q(I_{t-1}|I_t, I_0^\theta, I_{res}^\theta), \tag{23}$$

where $I_0^\theta = I_t - \bar{\alpha}_t I_{res}^\theta - \bar{\beta}_t \epsilon_\theta$ by Eq. 5. Here we only consider $L_{t-1}$ in (Ho et al., 2020),

$$L_{t-1} = D_{KL}(q(I_{t-1}|I_t, I_0, I_{res})||p_\theta(I_{t-1}|I_t)) \tag{24}$$

$$= \mathbb{E}\left[\left\|I_t - \alpha_t I_{res} - \frac{\beta_t^2}{\bar{\beta}_t}\epsilon - (I_t - \alpha_t I_{res}^\theta - \frac{\beta_t^2}{\bar{\beta}_t}\epsilon_\theta)\right\|^2\right], \tag{25}$$

where $D_{KL}$ denotes KL divergence. Ignoring the coefficients and the cross term $< I_{res} - I_{res}^\theta, \epsilon - \epsilon_\theta >$, we obtain the following simplified training objective,

$$L_{res}(\theta) + L_\epsilon(\theta), \tag{26}$$

where (repeat Eq. 11 here)

$$L_{res}(\theta) := \mathbb{E}\left[\lambda_{res}\left\|I_{res} - I_{res}^\theta(I_t, t, I_{in})\right\|^2\right], L_\epsilon(\theta) := \mathbb{E}\left[\lambda_\epsilon\left\|\epsilon - \epsilon_\theta(I_t, t, I_{in})\right\|^2\right], \tag{27}$$

and $\lambda_{res}, \lambda_\epsilon \in \{0, 1\}$.

## A.2 DETERMINISTIC IMPLICIT SAMPLING

If $q_\sigma(I_{t-1}|I_t, I_0, I_{res})$ is defined in Eq. 8, we have:

$$q(I_t|I_0, I_{res}) = \mathcal{N}(I_t; I_0 + \bar{\alpha}_t I_{res}, \bar{\beta}_t^2 \mathbf{I}). \tag{28}$$

*Proof.* Similar to the evolution from DDPM (Ho et al., 2020) to DDIM (Song et al., 2021a), we can prove the statement with an induction argument for $t$ from $T$ to 1. Assuming that Eq. 28 holds at $T$, we just need to verify $q(I_{t-1}|I_0, I_{res}) = \mathcal{N}(I_{t-1}; I_0 + \bar{\alpha}_{t-1} I_{res}, \bar{\beta}_{t-1}^2 \mathbf{I})$ at $t-1$ from $q(I_t|I_0, I_{res})$ at $t$ using Eq. 28. Given:

$$q(I_t|I_0, I_{res}) = \mathcal{N}(I_t; I_0 + \bar{\alpha}_t I_{res}, \bar{\beta}_t^2 \mathbf{I}), \tag{29}$$

$$q_\sigma(I_{t-1}|I_t, I_0, I_{res}) = \mathcal{N}(I_{t-1}; I_0 + \bar{\alpha}_{t-1} I_{res} + \sqrt{\bar{\beta}_{t-1}^2 - \sigma_t^2}\frac{I_t - (I_0 + \bar{\alpha}_t I_{res})}{\bar{\beta}_t}, \sigma_t^2 \mathbf{I}), \tag{30}$$

$$q(I_{t-1}|I_0, I_{res}) := \mathcal{N}(\tilde{\mu}_{t-1}, \tilde{\Sigma}_{t-1}) \tag{31}$$

Similar to obtaining $p(y)$ from $p(x)$ and $p(y|x)$ using Eq.2.113-Eq.2.115 in (Bishop & Nasrabadi, 2006), the values of $\tilde{\mu}_{t-1}$ and $\tilde{\Sigma}_{t-1}$ are derived as following:

$$\tilde{\mu}_{t-1} = I_0 + \bar{\alpha}_{t-1} I_{res} + \sqrt{\bar{\beta}_{t-1}^2 - \sigma_t^2}\frac{(I_0 + \bar{\alpha}_t I_{res}) - (I_0 + \bar{\alpha}_t I_{res})}{\bar{\beta}_t} = I_0 + \bar{\alpha}_{t-1} I_{res}, \tag{32}$$

$$\tilde{\Sigma}_{t-1} = \sigma_t^2 \mathbf{I} + (\frac{\sqrt{\bar{\beta}_{t-1}^2 - \sigma_t^2}}{\bar{\beta}_t})^2 \bar{\beta}_t^2 \mathbf{I} = \beta_{t-1}^2 \mathbf{I}. \tag{33}$$

Therefore, $q(I_{t-1}|I_0, I_{res}) = \mathcal{N}(I_{t-1}; I_0 + \bar{\alpha}_{t-1} I_{res}, \bar{\beta}_{t-1}^2 \mathbf{I})$. In fact, the case $(t = T)$ already holds, thus Eq. 28 holds for all $t$.

**Simplifying Eq. 8.** Eq. 8 can also be written as:

$$I_{t-1} = I_0 + \bar{\alpha}_{t-1} I_{res} + \sqrt{\bar{\beta}_{t-1}^2 - \sigma_t^2}\frac{I_t - (I_0 + \bar{\alpha}_t I_{res})}{\bar{\beta}_t} + \sigma_t \epsilon_t, \tag{34}$$

$$= \frac{\sqrt{\bar{\beta}_{t-1}^2 - \sigma_t^2}}{\bar{\beta}_t} I_t + (1 - \frac{\sqrt{\bar{\beta}_{t-1}^2 - \sigma_t^2}}{\bar{\beta}_t})I_0 + (\bar{\alpha}_{t-1} - \frac{\sqrt{\bar{\alpha}_t \bar{\beta}_{t-1}^2 - \sigma_t^2}}{\bar{\beta}_t})I_{res} + \sigma_t \epsilon_t \tag{35}$$

$$= I_t - (\bar{\alpha}_t - \bar{\alpha}_{t-1})I_{res} - (\bar{\beta}_t - \sqrt{\bar{\beta}_{t-1}^2 - \sigma_t^2})\epsilon + \sigma_t \epsilon_t, \tag{36}$$

where $\epsilon_t \sim \mathcal{N}(\mathbf{0}, \mathbf{I})$. Eq. 36 is consistent with Eq. 9, and Eq. 5 is used for the derivation from Eq. 35 to Eq. 36.

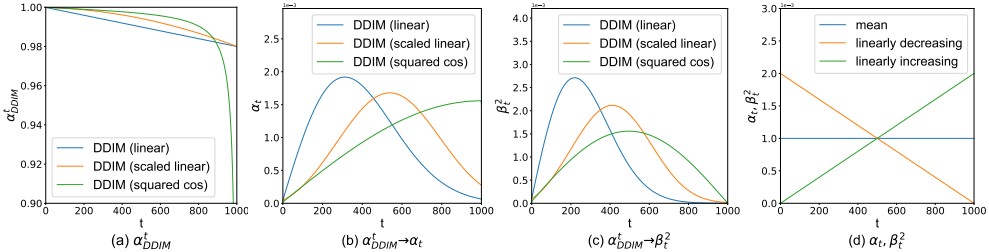

Figure 6: Coefficient transformation from DDIM (Song et al., 2021a) to RDDM using Eq. 13. (a) We show several schedules for $\alpha^t_{DDIM}$, e.g., linear (Song et al., 2021a), scaled linear (Rombach et al., 2022), and squared cosine (Nichol & Dhariwal, 2021). (b) We transform $\alpha^t_{DDIM}$ into $\alpha_t$ in our RDDM. (c) We transform $\alpha^t_{DDIM}$ into $\beta^2_t$ in our RDDM. (d) A few simple schedules. Using Eq. 14, "mean", "linearly increasing", and "linearly decreasing" can be denoted as $P(x, 0)$, $P(x, 1)$ and $P(1 - x, 1)$, respectively. See Algorithm 1 in Appendix A.3 for more details of (b) and (c).

### A.3 COEFFICIENT TRANSFORMATION

For image generation, $I_{in} = 0$, thus Eq. 12 can also be written as:

$$I_t = (\bar{\alpha}_t - 1)I_{res} + \bar{\beta}_t\epsilon \tag{37}$$

$$I_{res} = \frac{I_t - \bar{\beta}_t\epsilon}{\bar{\alpha}_t - 1}. \tag{38}$$

If the residuals $I^\theta_{res}$ are represented as a transformation of $\epsilon_\theta$ using Eq. 38, Eq. 9 is simplified to

$$I_{t-1} = I_t - (\bar{\alpha}_t - \bar{\alpha}_{t-1})I^\theta_{res} - (\bar{\beta}_t - \sqrt{\bar{\beta}^2_{t-1} - \sigma^2_t})\epsilon_\theta + \sigma_t\epsilon_t \tag{39}$$

$$= I_t - (\bar{\alpha}_t - \bar{\alpha}_{t-1})\frac{I_t - \bar{\beta}_t\epsilon_\theta}{\bar{\alpha}_t - 1} - (\bar{\beta}_t - \sqrt{\bar{\beta}^2_{t-1} - \sigma^2_t})\epsilon_\theta + \sigma_t\epsilon_t \tag{40}$$

$$= \frac{1 - \bar{\alpha}_{t-1}}{1 - \bar{\alpha}_t}I_t - (\frac{1 - \bar{\alpha}_{t-1}}{1 - \bar{\alpha}_t}\bar{\beta}_t - \sqrt{\bar{\beta}^2_{t-1} - \sigma^2_t})\epsilon_\theta + \sigma_t\epsilon_t \tag{41}$$

$$= \frac{\sqrt{\bar{\alpha}^{t-1}_{DDIM}}}{\sqrt{\bar{\alpha}^t_{DDIM}}}I_t - (\frac{\sqrt{\bar{\alpha}^{t-1}_{DDIM}}\sqrt{1 - \bar{\alpha}^t_{DDIM}}}{\sqrt{\bar{\alpha}^t_{DDIM}}} - \sqrt{1 - \bar{\alpha}^{t-1}_{DDIM} - \sigma^2_t})\epsilon_\theta + \sigma_t\epsilon_t \tag{42}$$

$$= \sqrt{\bar{\alpha}^{t-1}_{DDIM}}\left(\frac{I_t - \sqrt{1 - \bar{\alpha}^t_{DDIM}}\epsilon_\theta}{\sqrt{\bar{\alpha}^t_{DDIM}}}\right) + \sqrt{1 - \bar{\alpha}^{t-1}_{DDIM} - \sigma^2_t}\epsilon_\theta + \sigma_t\epsilon_t. \tag{43}$$

Eq. 43 is consistent with Eq.12 in DDIM (Song et al., 2021a) by replacing $\sigma^2_t$ with $\sigma^2_t(DDIM)$, and Eq. 13 is used for the derivation from Eq. 41 to Eq. 42. Thus, our sampling process is consistent with that of DDPM (Song et al., 2021a) and DDIM (Ho et al., 2020) by transforming coefficient schedules.

We present the pipeline of coefficient transformation in Algorithm 1. Fig. 6 shows the result of coefficient transformation. In Eq. 13, in addition to the coefficient $\bar{\alpha}_t, \bar{\beta}^2_t$ being replaced by $\bar{\alpha}^t_{DDIM}$, the variance $\sigma^2_t$ is also replaced with $\sigma^2_t(DDIM)$ to be consistent with DDIM (Song et al., 2021a) ($\eta = 0$) and DDPM (Ho et al., 2020) ($\eta = 1$). In fact, for DDIM (Song et al., 2021a) ($\eta = 0$), the variance is equal to 0 and does not need to be converted. Therefore, we analyze the difference between the variance of our RDDM and the variance of DDPM (Ho et al., 2020) in Appendix A.4.

### A.4 PERTURBED GENERATION PROCESS WITH SUM-CONSTRAINED VARIANCE

From Eq. 9, the variance of our RDDM ($\eta = 1$) is

$$\sigma^2_t(RDDM) = \eta\frac{\beta^2_t\bar{\beta}^2_{t-1}}{\bar{\beta}^2_t}. \tag{44}$$

We replace $\bar{\beta}^2_t$ by $\bar{\alpha}^t_{DDIM}$ using Eq. 13 and replace $\beta^2_t$ by $\bar{\beta}^2_t - \bar{\beta}^2_{t-1}$,

$$\sigma^2_t(RDDM) = \eta\bar{\alpha}^{t-1}_{DDIM}\frac{(1 - \bar{\alpha}^{t-1}_{DDIM})}{1 - \bar{\alpha}^t_{DDIM}}(1 - \frac{\bar{\alpha}^t_{DDIM}}{\bar{\alpha}^{t-1}_{DDIM}}), \tag{45}$$

---

**Algorithm 1:** Coefficient initialization, transformation, and adjustment.

---

**Input** : The initial conditions $\bar{\alpha}_T = 1$, $\bar{\beta}_T^2 > 0$, $T = 1000$, and $t \in \{1, 2, \dots T\}$. The hyperparameter $\eta = 1$ for the random generation process and $\eta = 0$ for deterministic implicit sampling. Variance modes have "DDIM" and "DDIM→RDDM". The coefficient adjustment mode Adjust="Alpha", "Beta", or "Alpha+Beta".

**Output:** The adjusted coefficients $\bar{\alpha}_t^*$, $\bar{\beta}_t^*$ and $\sigma_t^*$.

```
// (a) Coefficient initialization of DDIM (Song et al., 2021a)
```
1    $\beta_{DDIM}^t \leftarrow$ Linspace $(0.0001, 0.02, T)$         ▷ linear schedule (Song et al., 2021a)

2    $\alpha_{DDIM}^t \leftarrow 1 - \beta_{DDIM}^t$

3    $\bar{\alpha}_{DDIM}^t \leftarrow$ Cumprod $(\alpha_{DDIM}^t)$                   ▷ cumulative multiplication

4    $\sigma_t^2(DDIM) \leftarrow \eta \frac{(1-\bar{\alpha}_{DDIM}^{t-1})}{1-\bar{\alpha}_{DDIM}^t}(1 - \frac{\bar{\alpha}_{DDIM}^t}{\bar{\alpha}_{DDIM}^{t-1}})$

```
// (b) Coefficient transformation from DDIM (Song et al.,
   2021a) to our RDDM
```
5    $\bar{\alpha}_t \leftarrow 1 - \sqrt{\bar{\alpha}_{DDIM}^t}$                               ▷ Eq. 13

6    $\bar{\beta}_t \leftarrow \sqrt{1 - \bar{\alpha}_{DDIM}^t}$                                ▷ Eq. 13

7    $\sigma_t^2(RDDM) \leftarrow \eta \frac{(\bar{\beta}_t^2 - \bar{\beta}_{t-1}^2)\bar{\beta}_{t-1}^2}{\bar{\beta}_t^2}$

```
// (c) Select variance schedule
```
8    **if** *Variance=="DDIM"* **then**

9      $\sigma_t^* \leftarrow \sqrt{\sigma_t^2(DDIM)}$    ▷ consistent sampling process with DDIM (Song et al., 2021a) and DDPM (Ho et al., 2020)

10   **else if** *Variance=="DDIM→RDDM"* **then**

11      $\sigma_t^* \leftarrow \sqrt{\sigma_t^2(RDDM)}$                       ▷ sum-constrained variance schedule

12   **end**

```
// (d) Coefficient adjustment
```
13   $\alpha_t \leftarrow$ Power $(1 - t/T, 1)$                  ▷ linearly decreasing by Eq. 14

14   $\beta_t^2 \leftarrow$ Power $(t/T, 1) \cdot \bar{\beta}_T^2$            ▷ control the noise intensity in $I_T$ by $\bar{\beta}_T^2$

15   **if** *Adjust=="Alpha"* **then**

16      $\bar{\alpha}_t^* \leftarrow$ Cumsum $(\alpha_t)$, $\bar{\beta}_t^* \leftarrow \bar{\beta}_t$                     ▷ cumulative sum

17   **else if** *Adjust=="Beta"* **then**

18      $\bar{\alpha}_t^* \leftarrow \bar{\alpha}_t$, $\bar{\beta}_t^* \leftarrow \sqrt{\text{Cumsum}(\beta_t^2)}$               ▷ cumulative sum

19   **else if** *Adjust=="Alpha+Beta"* **then**

20      $\bar{\alpha}_t^* \leftarrow$ Cumsum $(\alpha_t)$, $\bar{\beta}_t^* \leftarrow \sqrt{\text{Cumsum}(\beta_t^2)}$      ▷ coefficient reinitialization

21   **else**

22      $\bar{\alpha}_t^* \leftarrow \bar{\alpha}_t$, $\bar{\beta}_t^* \leftarrow \bar{\beta}_t$

23   **end**

24   return $\bar{\alpha}_t^*, \bar{\beta}_t^*, \sigma_t^*$                ▷ sampling with adjusted coefficients by Eq. 9

---

while the variance of DDPM (Ho et al., 2020) ($\eta = 1$) is

$$\sigma_t^2(DDIM) = \eta \frac{(1 - \bar{\alpha}_{DDIM}^{t-1})}{1 - \bar{\alpha}_{DDIM}^t}(1 - \frac{\bar{\alpha}_{DDIM}^t}{\bar{\alpha}_{DDIM}^{t-1}}). \tag{46}$$

Our variance is much smaller than the variance of DDPM (Ho et al., 2020) because $0 < \bar{\alpha}_{DDIM}^{t-1} < \alpha_{DDIM}^1 < 1$ (e.g., $\alpha_{DDIM}^1 = 0.02$ in linear schedule (Song et al., 2021a)). Compared to $\sigma_t^2(DDIM) \approx 1$ (Song et al., 2021b), the variance of our RDDM is sum-constrained,

$$\sum_{i=1}^T \sigma_t^2(RDDM) = \sum_{i=1}^T \eta\beta_t^2 \frac{\bar{\beta}_{t-1}^2}{\bar{\beta}_t^2} \leq \sum_{i=1}^T \beta_t^2 \leq 1, \tag{47}$$

where $\sum_{i=1}^T \beta_t^2 = \bar{\beta}_T^2 = 1$ for image generation. This is also consistent with the directional residual diffusion process with perturbation defined in Eq. 5. A qualitative comparison of our RDDM ($\eta = 1$) with DDIM (Song et al., 2021a) ($\eta = 0$) and DDPM (Ho et al., 2020) ($\eta = 1$) is shown in Fig. 7. Notably, for $\eta = 0$, our RDDM is consistent with DDIM (Song et al., 2021a) (in Fig. 2(a)(b)).

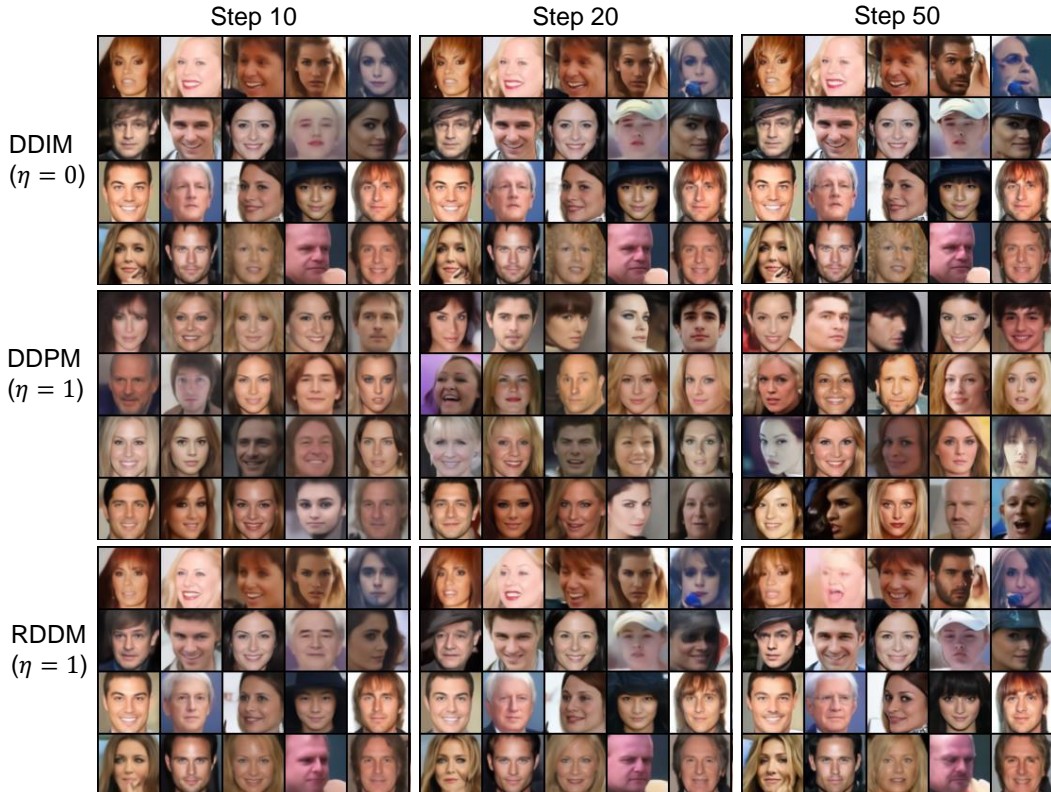

Figure 7: Perturbed generation process with sum-constrained variance on the CelebA $64 \times 64$ dataset (Liu et al., 2015). When the coefficients $(\bar{\alpha}_t, \bar{\beta}_t)$ are the same, RDDM ($\eta = 0$) is the same as DDIM ($\eta = 0$), but RDDM ($\eta = 1$) is different from DDPM ($\eta = 1$) due to different variances ($\sigma_t^2$). At different sampling steps, our RDDM has the same total noise, while ddpm has a different total noise. Notably, all the results in Fig. 7 can be generated from the same pre-trained model via variance transformation in Appendix A.3. In other words, the RDDM provides a sum-constrained variance strategy, which can be used directly in the pre-trained DDPM without re-training the model.

## A.5   COMPARISON WITH OTHER METHODS

The main difference is that to adapt the denoising diffusion, score, flow, or Schrödinger's bridge to image restoration, they choose the noise (Shadow Diffusion (Guo et al., 2023), SR3 (Saharia et al., 2022), and WeatherDiffusion (Özdenizci & Legenstein, 2023)), the residual (DvSR (Whang et al., 2022) and Rectified Flow (Liu et al., 2023d)), the target image (ColdDiffusion (Bansal et al., 2022), InDI (Delbracio & Milanfar, 2023), and consistency models (Song et al., 2023)), **or** its linear transformation term (I2SB (Liu et al., 2023a)), which is similar to a special case of our RDDM when it only predicts noise (SM-N) or residuals (SM-Res), while we introduce residual estimation but also embrace noise both for generation and restoration (SM-Res-N). We highlight that residuals and noise are equally important, e.g., the residual prioritizes certainty while the noise emphasizes diversity.

**Differences from DDPM (Ho et al., 2020).** 1) DDPM is not interpretable for image restoration, while our RDDM is a unified, interpretable diffusion model for both image generation and restoration. 2) Differences in the definition of the forward process lead to different variance strategies. Our RDDM has sum-constrained variance (much smaller than the variance of DDPM), while DDPM has preserving variance (Song et al., 2021b), as shown in Fig. 2(a)(b).

**Differences from IDDPM (Nichol & Dhariwal, 2021).** In the original DDPM (Ho et al., 2020), for the transfer probabilities $p_\theta(I_{t-1}|I_t)$ in Eq. 3, the mean $\mu_\theta(I_t, t)$ is learnable, while the variance $\Sigma_t$ is fixed. IDDPM (Nichol & Dhariwal, 2021) highlights the importance of **estimating both the mean and variance**, demonstrating that learning variances allow for fewer sampling steps with negligible differences in sample quality. However, IDDPM (Nichol & Dhariwal, 2021) still only

involves denoising procedures, and crucially, IDDPM, like DDPM, is thus not interpretable for image restoration. In addition, IDDPM (Nichol & Dhariwal, 2021) proposes three alternative ways to parameterize $\mu_\theta(I_t, t)$, i.e., predict mean $\mu_\theta(I_t, t)$ directly with a neural network, predict noise $\epsilon$, or predict clean image $I_0$. IDDPM (Nichol & Dhariwal, 2021) does not predict the clean image $I_0$ and noise $\epsilon$ at the same time, while both the residuals and the noise are predicted for our SM-Res-N.

The **essential difference** is that, our RDDM contains a mixture of three terms (i.e., input images $I_{in}$, target images $I_0$, and noise $\epsilon$) beyond DDPM/IDDPM (a mixture of two terms, i.e, $I_0$ and $\epsilon$). **We emphasize that residuals and noise are equally important**: the residual prioritizes certainty, while the noise emphasizes diversity. Furthermore, our RDDM preserves the original DDPM generation framework by coefficient transformation (Eq. 13), enabling seamless transfer of improvement techniques from DDPM, such as variance optimization from IDDPM.

**Differences from ColdDiffusion (Bansal et al., 2022).** 1) ColdDiffusion aims to remove the random noise entirely from the diffusion model, and replace it with other transforms (e.g., blur, masking), while our RDDM still embraces noise diffusion. Notably, we argue that noise is necessary for generative tasks that emphasize diversity (see Table 1). In fact, since ColdDiffusion discards random noise, extra noise injection is required to improve generation diversity. 2) To simulate the degradation process for different restoration tasks, ColdDiffusion attempts to use a Gaussian blur operation for deblurring, a snowification transform for snow removal., etc. These explorations may lose generality and differ fundamentally from our residual learning. RDDM represents directional diffusion from target images to input images using residuals, without designing specific degradation operators for each task. Additionally, RDDM provides solid theoretical derivation, while ColdDiffusion lacks theoretical justification.

**Differences from DvSR (Whang et al., 2022).** Whang et al. (2022) indeed use residual. But they 1) predict the initial clean image from a blurring image via a traditional (non-diffusion) network, calculate the residuals between the ground truth of the clean image and the predicted clean image 2) use denoising-based diffusion models predict noise like DDPM (Ho et al., 2020) and use a linear transformation of the noise to represent the residuals. They treat the residual predictions as an image generation task, aiming to produce diverse and plausible outputs based on the initial predicted clean image. Beyond simply building a diffusion model on top of residuals, we redefine a new forward process that allows simultaneous diffusion of residuals and noise, wherein the target image progressively diffuses into a purely noise or a noise-carrying input image.

**Differences from InDI (Delbracio & Milanfar, 2023) and I2SB (Liu et al., 2023a).** We can conclude that the forward diffusion of InDI, I2SB, and our RDDM is consistent in the form of a mixture of three terms (i.e., input images $I_{in}$, target images $I_0$, and noise $\epsilon$) beyond the denoising-based diffusion (a mixture of two terms, i.e, $I_0$ and $\epsilon$). Substituting $I_{res} = I_{in} - I_0$ into Eq. 12 results in $I_t = \bar{\alpha}_t I_{in} + (1 - \bar{\alpha}_t)I_0 + \bar{\beta}_t \epsilon$. This resulted $I_t$ has the same format as Eq.8 in InD ($x_t = ty + (1-t)x + \sqrt{t}\epsilon_t \eta_t$), and is the same format as Eq.11 in I2SB. Similar to Eq. 13 (from our RDDM to DDPM/DDIM), transforming coefficients leads to complete consistency. However, our RDDM can further extend DDPM/DDIM, InD, and I2SB to independent double diffusion processes, and holds the potential to pave the way for the multi-dimensional diffusion process. From the initial stages of constructing a new forward process, our RDDM uses independent coefficient schedules to control the diffusion of residuals and noise. This provides a more general, flexible, and scalable framework, and inspires our partially path-independent generation process, demonstrated in Fig. 3 and Fig. 15(b-f) with stable generation across various diffusion rates and path variations.

# B EXPERIMENTS

## B.1 TRAINING DETAILS

We use a UNet architecture[9] for both residual prediction and noise prediction in our RDDM. The UNet settings remain consistent across all tasks, including the channel size (64) and channel multiplier (1,2,4,8). Detailed experimental settings can be found in Table 4. Training and testing for all experiments in Table 4 can be conducted on a single Nvidia GTX 3090.

---

[9]Our RDDM is implemented by modifying https://github.com/lucidrains/denoising-diffusion-pytorch repository.

Table 4: Experimental settings for training our RDDM."SM-Res-N-2Net" is described in Appendix B.2. Two optimizers can be implemented in $L_{res}, L_\epsilon$.

| Tasks | Image Restoration | | | | Image Generation | Image Inpainting | Image Translation |
|---|---|---|---|---|---|---|---|
| | Shadow Removal | Low-light | Deblurring | Deraining | | | |
| Datasets | ISTD | LOL SID-RGB | GoPro | RainDrop | CelebA | CelebA-HQ | CelebA-HQ AFHQ |
| Batch size | 1 | 1 | 1 | 1 | 128 | 64 | 64 |
| Image size | 256 | 256 | 256 | 256 | 64 | 64 | 64 |
| $\bar{\beta}_T^2$ | 0.01 | 1 | 0.01 | 1 | 1 | 1 | 1 |
| $I_{in}$ | $I_{in}$ | $I_{in}$ | $I_{in}$ | $I_{in}$ | 0 | 0 | 0 |
| Sampling steps | 5 | 2 | 2 | 5 | 10 | 10 | 10 |
| Loss type | $\ell_1$ | $\ell_1$ | $\ell_1$ | $\ell_1$ | $\ell_2$ | $\ell_2$ | $\ell_2$ |
| Loss | $L_{res}+L_\epsilon$ | $L_{res}$ | $L_{res}+L_\epsilon$ | $L_{res}+L_\epsilon$ | $L_\epsilon$ | $L_{res}, L_\epsilon$ | $L_{res}, L_\epsilon$ |
| Sampling Method | SM-Res-N-2Net | SM-Res | SM-Res-N-2Net | SM-Res-N-2Net | SM-N | SM-Res-N-2Net | SM-Res-N-2Net |
| Optimizer | Adam | Adam | Adam | Adam | RAdam | RAdam | RAdam |
| Learning rate | 8e-5 | 8e-5 | 8e-5 | 8e-5 | 2e-4 | 2e-4 | 2e-4 |
| Training iterations | 80k | 80k | 400k | 120k | 100k | 100k | 100k |
| Schedules | $\alpha_t : P(1-x,1)$ $\beta_t^2 : P(x,1)$ | $\alpha_t : P(1-x,1)$ $\beta_t^2 : P(x,1)$ | $\alpha_t : P(1-x,1)$ $\beta_t^2 : P(x,1)$ | $\alpha_t : P(1-x,1)$ $\beta_t^2 : P(x,1)$ | $\alpha_{DDIM}^t \to$ $\alpha_t, \beta_t^2$ | $\alpha_t : P(1-x,1)$ $\beta_t^2 : P(x,1)$ | $\alpha_{DDIM}^t \to$ $\alpha_t, \beta_t^2$ |

**Image Generation.** For comparison with DDIM Song et al. (2021a), we convert the $\alpha_{DDIM}^t$ schedule of DDIM Song et al. (2021a) into the $\alpha_t$ and $\beta_t^2$ schedules of our RDDM using Eq. 13 in Section 4.2 and Section 5. In fact, a better coefficient schedule can be used in our RDDM, e.g., $\alpha_t$ (linearly decreasing) and $\beta_t^2$ (linearly increasing) in Table 2.

**Image Restoration.** We extensively evaluate our method on several image restoration tasks, including shadow removal, low-light enhancement, image deraining, and image deblurring on 5 different datasets. For fair comparisons, the results of other SOTA methods are provided from the original papers whenever possible. For all image restoration tasks, the images are resized to 256, and the networks are trained with a batch size of 1. We use shadow masks and shadow images as conditions for shadow removal (similar to (Le & Samaras, 2019; Zhu et al., 2022a)), while other image restoration tasks use the degraded image as condition inputs. For low-light enhancement, we use histogram equalization for pre-processing. To cope with the varying tasks and dataset sizes, we only modified the number of training iterations, $\bar{\beta}_T^2$ and sampling steps (5 steps for shadow removal and deraining, 2 steps for low-light and deblurring) as shown in Table 4. $\alpha_t$ is initialized using a linearly decreasing schedule (i.e., $P(1-x,1)$ in Eq. 14), while $\beta_t^2$ is initialized using a linearly decreasing schedule (i.e., $P(x,1)$). The quantitative results were evaluated by the peak signal to noise ratio (PSNR), structural similarity (SSIM), and learned perceptual image patch similarity (LPIPS) (Zhang et al., 2018).

Notably, our RDDM uses an identical UNet architecture and is trained with a batch size of 1 for all these tasks. In contrast, SOAT methods often involve elaborate network architectures, such as multi-stage (Fu et al., 2021; Zamir et al., 2021; Zhu et al., 2022b), multi-branch (Cun et al., 2020), and GAN (Wang et al., 2018; Kupyn et al., 2019; Qian et al., 2018), or sophisticated loss functions like the chromaticity (Jin et al., 2021), texture similarity (Zhang et al., 2019), and edge loss (Zamir et al., 2021).

**Image Inpainting and Image Translation.** To increase the diversity of the generated images, conditional input images were not fed into the deresidual and denoising network (see Fig. 18).

## B.2 SAMPLING DETAILS

**SM-Res or SM-N.** We present the motivation, conceptualization, and implementation pipeline (Algorithm 2) of the **Automatic Target Domain Prediction Algorithm (ATDP)** as follows:

Step 1. At the initial simultaneous training (similar to SM-Res-N), we do not know whether the network output ($I_{out}$) is residual or noise. Therefore, we set $\lambda_{res}^\theta = 0.5$ to denote the probability that the output is residual ($I_{res}^\theta$), and $1 - \lambda_{res}^\theta$ is the probability that the output is noise ($\epsilon_\theta$).

Step 2. We then impose loss constraints on both residual and noise estimation weighted by the learned parameter ($\lambda_{res}^\theta$), as follows:

$$L_{auto}(\theta) := \lambda_{res}^\theta E\left[\left\|I_{res} - I_{res}^\theta(I_t, t, I_{in})\right\|^2\right] + (1 - \lambda_{res}^\theta)E\left[\left\|\epsilon - \epsilon_\theta(I_t, t, I_{in})\right\|^2\right]. \quad (48)$$

---

**Algorithm 2:** Training Pipeline Using ATDP.

---

**Input** : A degraded input image $I_{in}$ and its corresponding ground truth image $I_0$. Gaussian noise $\epsilon$. Time condition $t$. Coefficient schedules $\bar{\alpha}$ and $\bar{\beta}_t$. The initial learnable parameters $\lambda^{\theta}_{res} = 0.5$. Network $G$ with parameters $\theta$. The initial learning rate $l$. $n$ is the training iterations number. $m$ is the iterations number of ATDP. The threshold of shifting training strategies, $\delta = 0.01$.

**Output:** Trained well parameters, $\theta, \lambda^{\theta}_{res}$.

---

1   $\theta \leftarrow$ InitWight $(G)$                  ▷ initialize network parameters

2   **for** $i \leftarrow 1$ **to** $n + m$ **do**

3      $t \sim$ Uniform $(\{1, 2, ..., T\}), \epsilon \sim \mathcal{N}(\mathbf{0}, \mathbf{I}), I_{res} \leftarrow I_{in} - I_0$

4      $I_t \leftarrow I_0 + \bar{\alpha}_t I_{res} + \bar{\beta}_t \epsilon$                ▷ synthesize $I_t$ by Eq. 5

5      $I_{out} \leftarrow G(I_t, t, I_{in})$

6      $I^{\theta}_{res} \leftarrow \lambda^{\theta}_{res} \times I_{out} + (1 - \lambda^{\theta}_{res}) \times f_{\epsilon \rightarrow res}(I_{out})$   ▷ $f_{\epsilon \rightarrow res}(\cdot)$: from $\epsilon$ to $I_{res}$ using Eq. 12

7      $\epsilon_{\theta} \leftarrow \lambda^{\theta}_{res} \times f_{res \rightarrow \epsilon}(I_{out}) + (1 - \lambda^{\theta}_{res}) \times I_{out}$    ▷ $f_{res \rightarrow \epsilon}(\cdot)$: from $I_{res}$ to $\epsilon$ using Eq. 12

8      $L_{auto} \leftarrow$ Loss $(I^{\theta}_{res}, I_{res}, \epsilon_{\theta}, \epsilon)$             ▷ based on Eq. 48

9      $\theta, \lambda^{\theta}_{res} \overset{+}{\leftarrow} -\nabla_{\theta, \lambda^{\theta}_{res}}(\mathcal{L}_{auto}, l)$           ▷ updating gradient

10      **if** abs $(\lambda^{\theta}_{res} - 0.5) < \delta$ **then**

11         pass                    ▷ adversarial-like training

12      **else**

13         $\lambda^{\theta}_{res} \leftarrow$ Detach $(\lambda^{\theta}_{res})$          ▷ halt the gradient updates

14         $\theta \leftarrow$ InitWight $(G)$         ▷ reinitialize network parameters

15         **if** $\lambda^{\theta}_{res} > 0.5$ **then**

16            $\lambda^{\theta}_{res} \leftarrow 1$                  ▷ SM-Res

17         **else**

18            $\lambda^{\theta}_{res} \leftarrow 0$                  ▷ SM-N

19         **end**

20      **end**

21 **end**

---

Table 5: Ablation studies of sampling methods and network structures. "SM-Res-N-1Net"+"one network" denotes to output 6 channels using a network, where the 0-3-th channels are residual and the 3-6-th channels are noise.

| Sampling Method | Network | MAE(↓) | SSIM(↑) | PSNR(↑) |
|---|---|---|---|---|
| SM-Res | Residual network | 4.76 | 0.959 | 30.72 |
| SM-Res-N-2Net | Residual network+noise network | 4.67 | 0.962 | 30.91 |
| SM-Res-N-1Net | One network, only shared encoder | 4.72 | 0.959 | 30.73 |
| SM-Res-N-1Net | One network | **4.57** | **0.963** | **31.10** |

The joint loss functions $L_{auto}(\theta)$ drive the network to gradually favor either residuals or noise based on the input. For example, in the image restoration task with deterministic input, it should be simpler for the network to estimate a clear image than noise. In contrast, for the image generation task with random noise input, it is simpler for the network to estimate the noise than a clear image.

Step 3. To enable learning of $\lambda^{\theta}_{res}$, we then include it in the network computation, allowing gradient transmission. Since $\lambda^{\theta}_{res}$ denotes the probability that the output is residual, the estimated residual $I^{\theta}_{res}$ can be represented as $\lambda^{\theta}_{res} \times I_{out} + (1 - \lambda^{\theta}_{res}) \times f_{\epsilon \rightarrow res}(I_{out})$. $f_{\epsilon \rightarrow res}(\cdot)$ represents the transformation from $\epsilon$ to $I_{res}$ using Eq. 12. Similarly, $\epsilon_{\theta}$ can be represented as $\lambda^{\theta}_{res} \times f_{res \rightarrow \epsilon}(I_{out}) + (1 - \lambda^{\theta}_{res}) \times I_{out}$. This is very similar to the cross-entropy loss function.

Step 4. As training approaches are completed, our objective should be to estimate only noise (SM-N) or residuals (SM-Res). By utilizing the learned $\lambda^{\theta}_{res}$, we can determine when to switch from an adversarial-like process (residuals vs. noise in Step 2) to a single prediction (residuals or noise). This transition can be controlled, for instance, by setting a condition such as abs($\lambda^{\theta}_{res} - 0.5) \geq 0.01$. When the network's tendency to estimate residuals surpasses 51% probability, we set $\lambda^{\theta}_{res}$ to 1 and halt the gradient updates for $\lambda^{\theta}_{res}$.

The experimental results were consistent with the empirical analysis in Section 3.3 and verified the effectiveness of ATDP. For instance, the initial simultaneous training switches to residual learning (SM-Res) for shadow removal and low-light enhancement in approximately 300 iterations, and to denoising learning (SM-N) for image generation in approximately 1000 iterations. To summarize, ATDP achieves the same inference cost as the current denoising-based diffusion methods (Ho et al., 2020) with the plug-and-play training strategy.

**SM-Res-N.** Both the residuals and the noise are predicted, which can be implemented with two or one networks. **SM-Res-N-2Net.** If computational resources are sufficient, two separate networks can be trained for noise and residual predictions, and the optimal sampling method can be determined during testing. This setting easily obtains a well-suited network for the target task, and facilitates the exploration of the decoupled diffusion process and the partially path-independent generation process in Section 4. **SM-Res-N-1Net.** To avoid training two separate networks, another solution is to simply use a joint network (i.e., a shared encoder and decoder) to output 6 channels where the 0-3-th channels are residual and the 3-6-th channels are noise. This setting loses the decoupling property of RDDM, but can achieve dual prediction with a slight cost. Table 5 shows that the joint network (i.e., SM-Res-N-1Net+One network) achieves the best shadow removal results (MAE 4.57), even better than two independent networks (4.67). A network with the shared encoder (MAE 4.72) has a slight performance degradation compared to the independent two networks (4.67).

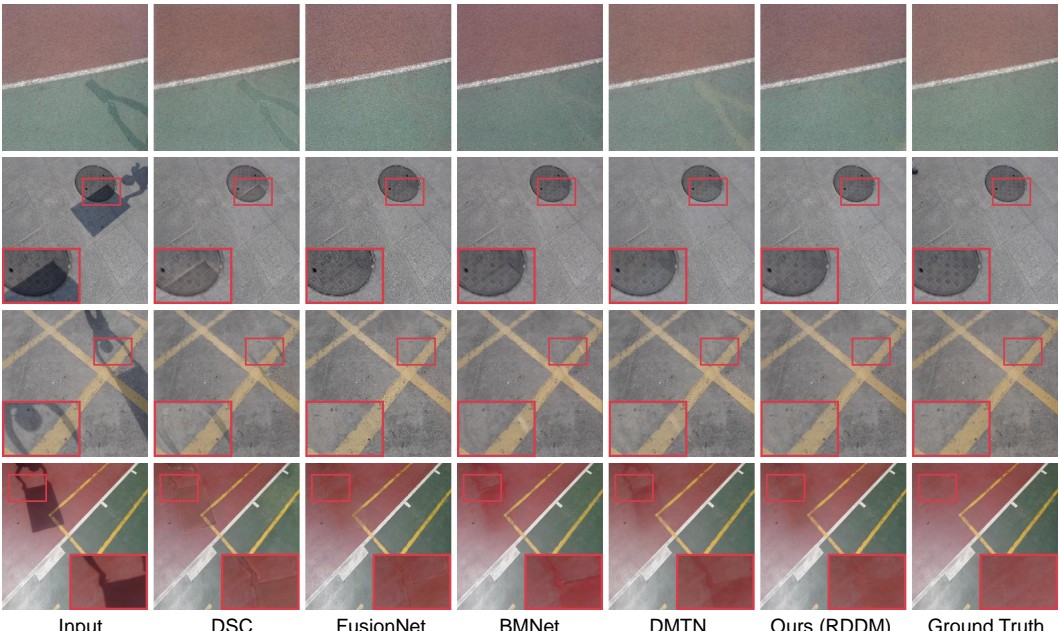

Figure 8: More visual comparison results for shadow removal on the ISTD dataset (Wang et al., 2018).

Table 6: Shadow removal results on the ISTD dataset (Wang et al., 2018). We report the MAE, SSIM and PSNR in the shadow area (S), non-shadow area (NS), and whole image (ALL).

| Method | MAE(↓) | | | SSIM(↑) | | | PSNR(↑) | | | LPIPS(↓) |
|---|---|---|---|---|---|---|---|---|---|---|
| | S | NS | ALL | S | NS | ALL | S | NS | ALL | |
| ST-CGAN (Wang et al., 2018) | 10.33 | 6.93 | 7.47 | 0.981 | 0.958 | 0.929 | 33.74 | 29.51 | 27.44 | - |
| DSC (Hu et al., 2020) ¶ | 9.48 | 6.14 | 6.67 | 0.967 | - | - | 33.45 | - | - | - |
| DHAN (Cun et al., 2020) | 8.14 | 6.04 | 6.37 | 0.983 | - | - | 34.50 | - | - | - |
| CANet (Chen et al., 2021) | 8.86 | 6.07 | 6.15 | - | - | - | - | - | - | - |
| LG-ShadowNet (Liu et al., 2021b) | 10.23 | 5.38 | 6.18 | 0.979 | 0.967 | 0.936 | 31.53 | 29.47 | 26.62 | - |
| FusionNet (Fu et al., 2021) | 7.77 | 5.56 | 5.92 | 0.975 | 0.880 | 0.945 | 34.71 | 28.61 | 27.19 | 0.1204 |
| UnfoldingNet (Zhu et al., 2022b) | 7.87 | 4.72 | 5.22 | 0.987 | _0.978_ | 0.960 | **36.95** | 31.54 | 29.85 | - |
| BMNet (Zhu et al., 2022a) | 7.60 | 4.59 | 5.02 | _0.988_ | 0.976 | 0.959 | 35.61 | 32.80 | 30.28 | 0.0377 |
| DMTN (Liu et al., 2023b) | _7.00_ | _4.28_ | _4.72_ | **0.990** | **0.979** | **0.965** | 35.83 | _33.01_ | _30.42_ | _0.0368_ |
| Ours (RDDM) | **6.67** | **4.27** | **4.67** | _0.988_ | **0.979** | _0.962_ | _36.74_ | **33.18** | **30.91** | **0.0305** |

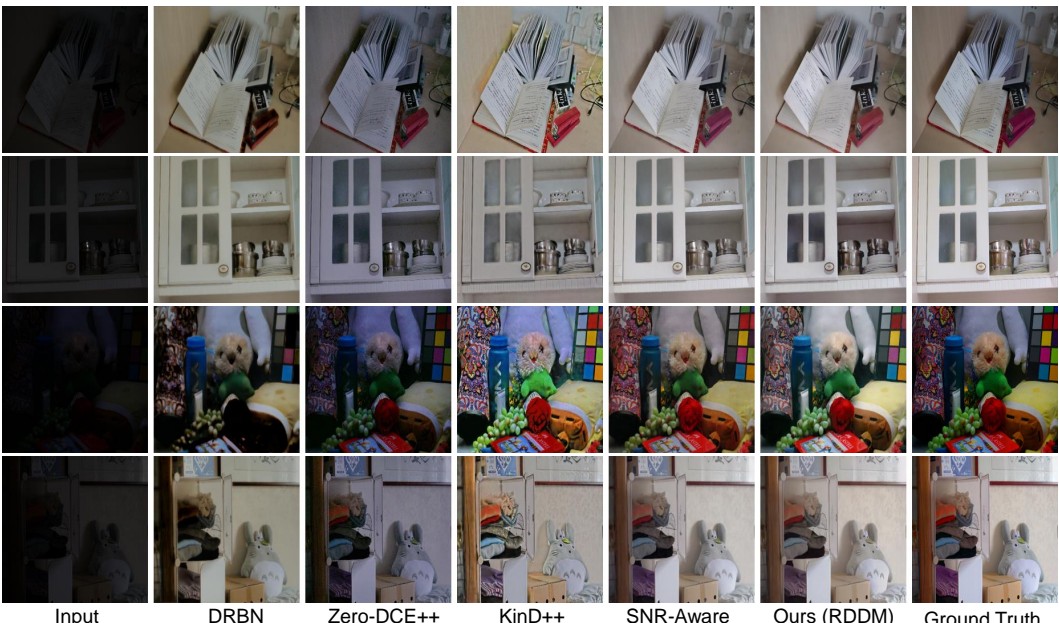

Input  DRBN  Zero-DCE++  KinD++  SNR-Aware  Ours (RDDM)  Ground Truth

Figure 9: More visual comparison results for Low-light enhancement on the LOL dataset (Wei et al., 2018).

Table 7: Quantitative comparison results of Low-light enhancement on the SID-RGB dataset (Xu et al., 2020) and deblurring on the GoPro dataset (Nah et al., 2017). The results of MIR-Net are reported by (Zheng et al., 2021).

| (a) Low-light (SID-RGB) | PSNR(↑) | SSIM(↑) | LPIPS(↓) | (b) Deblurring (GoPro) | PSNR(↑) | SSIM(↑) | LPIPS(↓) |
|---|---|---|---|---|---|---|---|
| SID (Chen et al., 2018) | 21.16 | 0.6398 | 0.4026 | Deblurgan-v2 (Kupyn et al., 2019) | 29.55 | 0.934 | 0.117 |
| D&E (Xu et al., 2020) | 22.13 | 0.7172 | 0.3794 | Suin *et al.* (Suin et al., 2020) | 31.85 | 0.948 | - |
| MIR-Net(Zamir et al., 2020; 2022) | 22.34 | 0.7031 | 0.3562 | MPRNet (Zamir et al., 2021) | 32.66 | 0.959 | 0.089 |
| UTVNet (Zheng et al., 2021) | 22.69 | 0.7179 | 0.3417 | DvSR (Whang et al., 2022) | 31.66 | 0.948 | 0.059 |
| SNR-Aware (Xu et al., 2022) | 22.87 | 0.625 | - | Uformer-B (Wang et al., 2022b) | 32.97 | 0.967 | 0.0089 |
| Our RDDM (2 step) | 23.97 | 0.8392 | 0.2433 | Our RDDM (2 step) | 32.40 | 0.963 | 0.0415 |
| Our RDDM (5 step) | 23.80 | 0.8378 | 0.2289 | Our RDDM (10 step) | 31.67 | 0.950 | 0.0379 |

### B.3 MORE RESULTS

**Shadow removal.** We compare RDDM with DSC (Hu et al., 2020), FusionNet (Fu et al., 2021), BMNet (Zhu et al., 2022a) and DMTN (Liu et al., 2023b) on the ISTD dataset (Wang et al., 2018). The ISTD dataset (Wang et al., 2018) contains shadow images, shadow masks, and shadow-free image triplets (1,330 for training; 540 for testing). Table 3(b), Fig. 4(b), and Fig. 8 demonstrate the superiority of our method. In addition, we compare RDDM with more shadow removal methods (e.g., ST-CGAN (Wang et al., 2018), DHAN (Cun et al., 2020), CANet (Chen et al., 2021), LG-ShadowNet (Liu et al., 2021b), UnfoldingNet (Zhu et al., 2022b)) in Table 6.

**Low-light enhancement.** We evaluate our RDDM on the LOL (Wei et al., 2018) (500 images) and SID-RGB (Xu et al., 2020) datasets (5,094 images), and compare our method with the current SOTA methods (Zhang et al., 2021; Liu et al., 2021a; Yuhui et al., 2023; Zhang et al., 2022; Xu et al., 2022; Zamir et al., 2022; Zheng et al., 2021). To unify and simplify the data loading pipeline for training, we only evaluate the RGB low-light image dataset (Wei et al., 2018; Xu et al., 2020), not the RAW datasets (e.g., FiveK (Bychkovsky et al., 2011)). Table 3(c), Fig. 4(c), and Fig. 9 show that our RDDM achieves the best SSIM and LPIPS (Zhang et al., 2018) and can recover better visual quality on the LOL (Wei et al., 2018) dataset. Table 7(a) shows the low-light enhancement results on the SID-RGB (Xu et al., 2020) dataset of different methods. Our RDDM outperforms the state-of-the-art SNR-Aware (Xu et al., 2022) by a **4.8**% PSNR and a **34.2**% SSIM improvement on the SID-RGB (Xu et al., 2020) dataset. Fig. 10 shows that our RDDM outperforms competitors in detail recovery (sharper text of 1st row), and color vibrancy (2nd & 3rd rows), avoiding issues like gray shading and detail blurring.

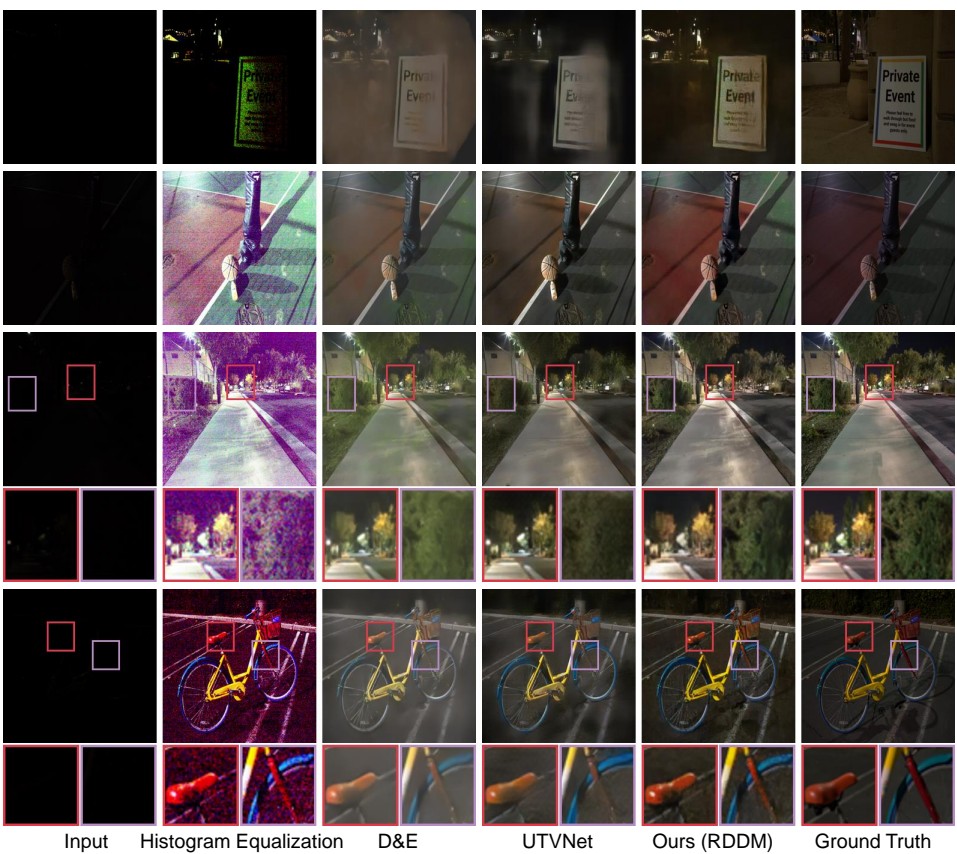

Figure 10: More visual comparison results for Low-light enhancement on the SID-RGB dataset (Xu et al., 2020).

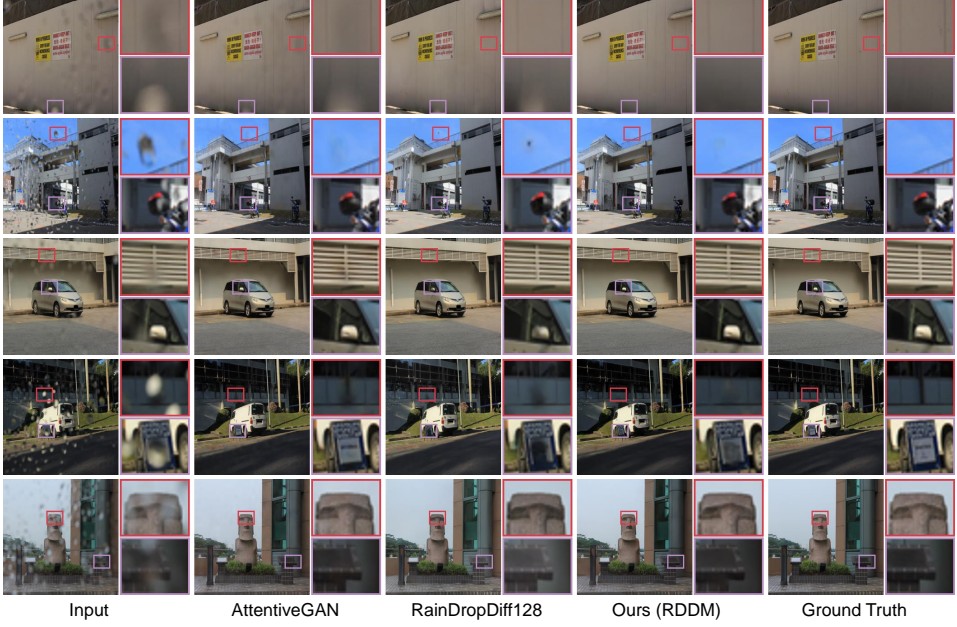

Figure 11: More visual comparison results for deraining on the RainDrop (Qian et al., 2018) dataset.

**Image deraining.** We make a fair comparison with the current SOTA diffusion-based image restoration method - RainDiff128 (Özdenizci & Legenstein, 2023) ("128" denotes the 128×128 patch size for training) on the RainDrop dataset (Qian et al., 2018) (1119 images). RainDiff128 (Özdenizci &

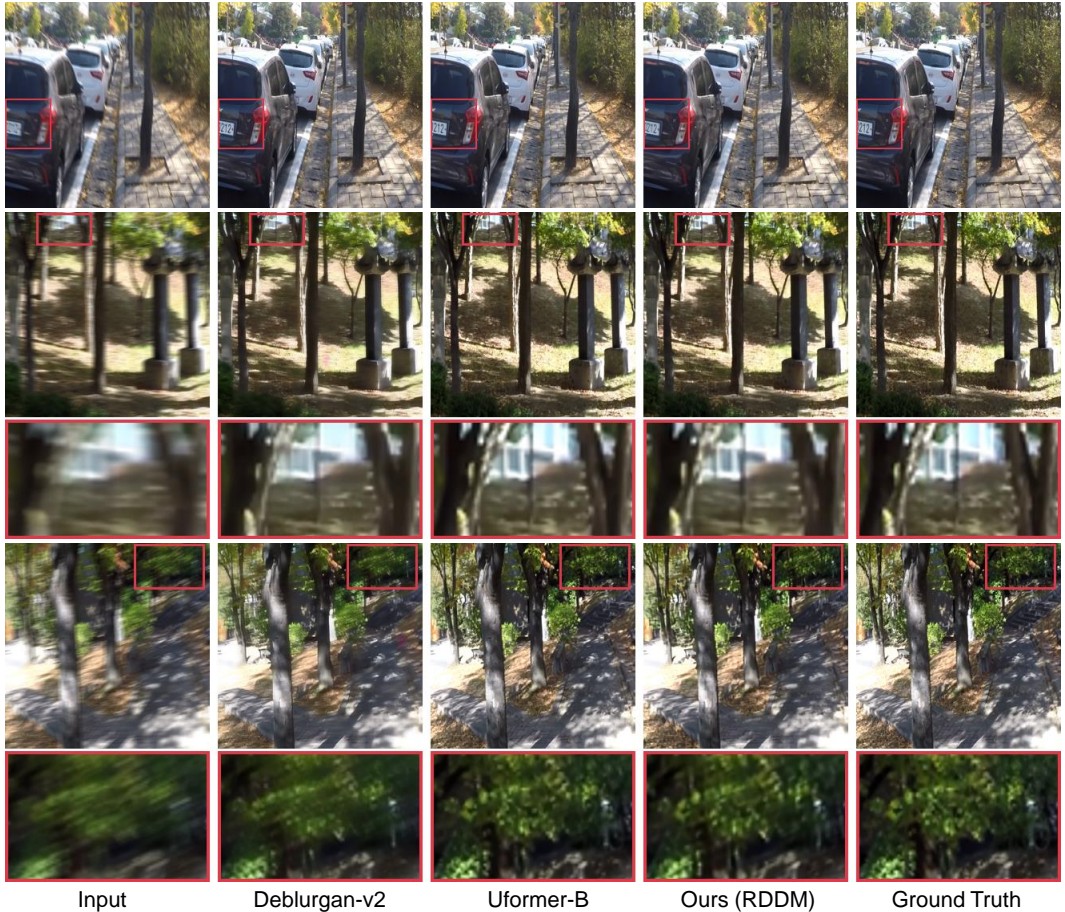

Figure 12: More visual comparison results for deblurring on the GoPro (Nah et al., 2017) dataset.

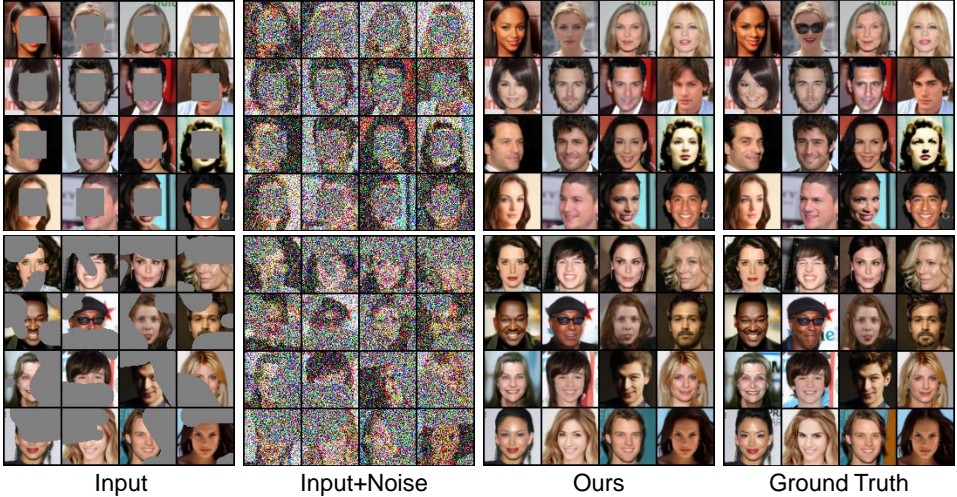

Figure 13: More visual results for image inpainting on the CelebA-HQ (Karras et al., 2018) dataset. The image resolution is resized to 64.

Legenstein, 2023) feeds the degraded input image as a condition to the denoising network, which requires 50 sampling steps to generate a clear image from the noise, while our RDDM requires only 5 sampling steps to recover the degraded image from the noise-carrying input image and outperforms RainDiff128 (Özdenizci & Legenstein, 2023), as shown in Table 3(d) and Fig. 11.

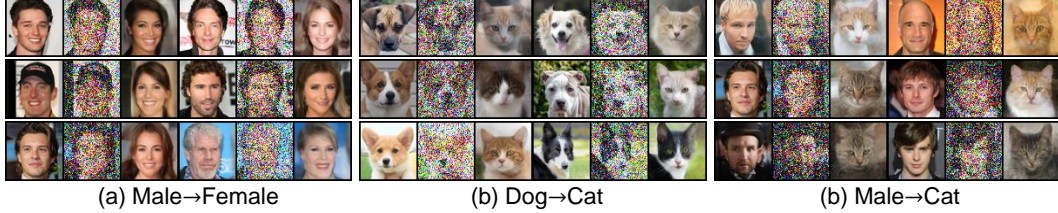

(a) Male→Female         (b) Dog→Cat         (b) Male→Cat

Figure 14: More visual results for image translation on the CelebA-HQ (Karras et al., 2018) and AFHQ (Choi et al., 2020) datasets. The image resolution is resized to 64.

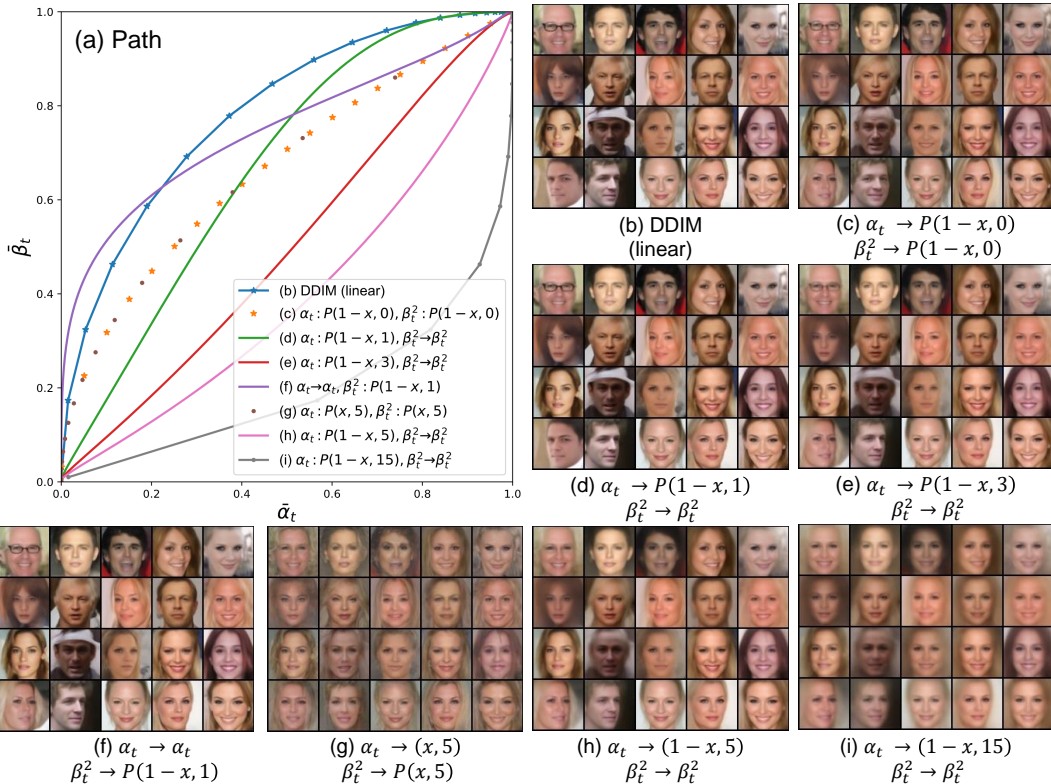

Figure 15: More visual results for the partially path-independent generation process. Two networks are used to estimate residuals and noise separately, i.e., $I^\theta_{res}(I_t, \bar{\alpha}_t \cdot T)$ and $\epsilon_\theta(I_t, \bar{\beta}_t \cdot T)$ ($\eta = 0$).

**Image deblurring.** We evaluate our method on the widely used deblurring dataset - GoPro (Nah et al., 2017) (3,214 images). Table 7(b) and Fig. 12 show that our method is competitive with the SOTA deblurring methods (e.g, MPRNet (Zamir et al., 2021), and Uformer-B (Wang et al., 2022b)).

**Image Inpainting and Image Translation.** We show more qualitative results of image inpainting (Fig. 13) and translation (Fig. 14(e)).

## B.4    PARTIALLY PATH-INDEPENDENT GENERATION PROCESS

Fig. 15(b-f) provides evidence supporting the partially path-independent generation process, demonstrating the robustness of the generative process within a certain range of diffusion rates (step size per step) and path variation, e.g., converting DDIM (Song et al., 2021a) to a uniform diffusion speed in Fig. 15(c). However, excessive disturbances can result in visual inconsistencies, as depicted in Fig. 15(h)(i). Furthermore, Fig. 15(c) and Fig. 15(g) illustrate that even when the paths are the same, the variation in diffusion speed significantly impacts the quality of the generated images. This highlights the importance of carefully considering and controlling the diffusion speed and generation path during the generation process.

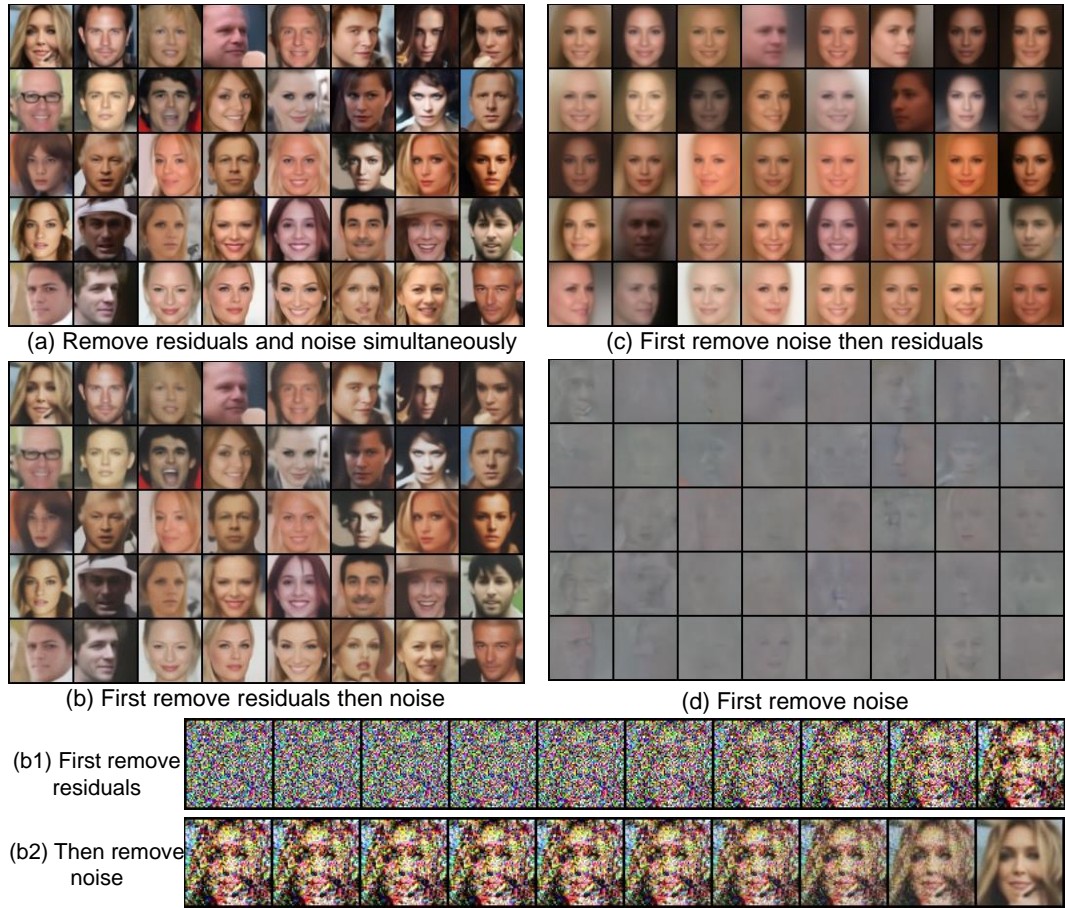

Figure 16: Special paths of the partially path-independent generation process. Two networks are used to estimate residuals and noise separately, i.e., $I_{res}^{\theta}(I_t, \bar{\alpha}_t \cdot T), \epsilon_{\theta}(I_t, \bar{\beta}_t \cdot T)$ ($\eta = 0$).

We also investigated two reverse paths to gain insight into the implications of the proposed partial path independence. In the first case, the residuals are removed first, followed by the noise: $I(T) \overset{-I_{res}}{\to} I(0) + \bar{\beta}_T \epsilon \overset{-\bar{\beta}_T \epsilon}{\to} I(0)$, as shown in Fig. 16(b1)(b2). The second case involves removing the noise first and then the residuals: $I(T) \overset{-\bar{\beta}_T \epsilon}{\to} I_{in} \overset{-I_{res}}{\to} I(0)$. In the first case, images are successfully generated (as shown in Fig. 16(b)) which exhibit a striking similarity to the default images in Fig. 16(a). However, the second case shown in Fig. 16(c) fails to go from $I_{in}$ to $I(0)$ due to $I_{in} = 0$ in the generation task. Figure 16(d) shows the intermediate visualization results of removing the noise first.

## B.5 ABLATION STUDIES

We have analyzed the sampling method in Table 1, the coefficient schedule in Table 2, and the network structure for SM-Res-N in Table 5.

**Sampling Methods.** We present the results for noise predictions only (SM-N) in Fig. 17. Fig. 17 (b) and (c) illustrate that estimating only the noise poses challenges as colors are distorted, and it becomes difficult to retain information from the input shadow image. We found that increasing sampling steps does not lead to improved results from Fig. 17 (b) to Fig. 17 (c), which may be an inherent limitation when estimating only the noise for image restoration. Actually, this is also reflected in DeS3 (Jin et al., 2022a) (a shadow removal method based on a denoising diffusion model), where DeS3 (Jin et al., 2022a) specifically designs the loss against color bias. Additionally, training with batch size 1 may contribute to poor results of only predicting noise. However, estimating only the residuals (SM-Res) with batch size 1 does not exhibit such problems for image restoration, as

demonstrated in Fig. 17 (d)&(e) and Table 1, further demonstrating the merits of our RDDM. For image inpainting, SM-Res-N can generate more realistic face images compared to SM-N and SM-Res, as shown in Fig. 18(d-f). If computational resources are sufficient, to obtain better image quality for an unknown task, we suggest that two separate networks can be trained for noise and residual predictions, and the optimal sampling method can be determined during testing. If computational resources are limited, the sampling method can be determined empirically (see Section 3.3).

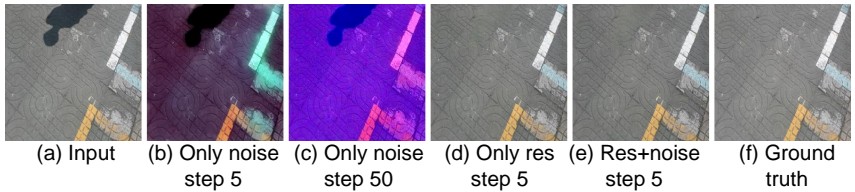

(a) Input    (b) Only noise step 5    (c) Only noise step 50    (d) Only res step 5    (e) Res+noise step 5    (f) Ground truth

Figure 17: Visualizing ablation studies of sampling methods.

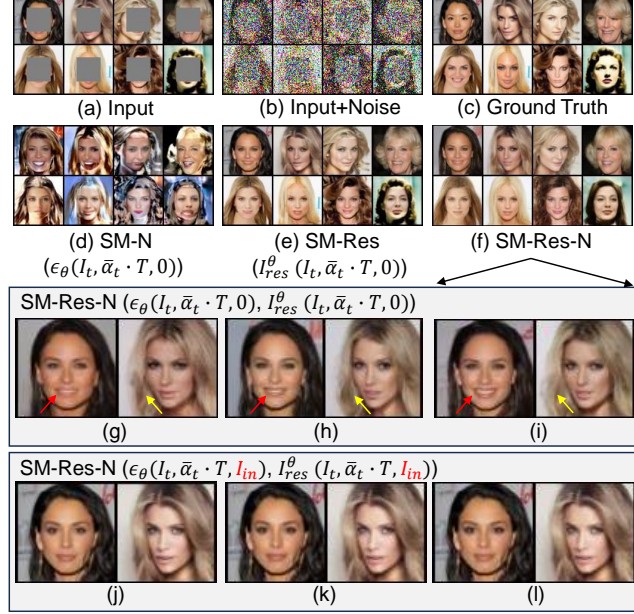

Figure 18: Visualizing ablation studies of sampling methods for image inpainting on the CelebA-HQ (Karras et al., 2018) dataset. (g-i) The conditional input image (a) is not used as an input to the deresidual and denoising network in the generation process from (b) to (f). Compared to (g-i), the diversity of the generated images in (j-l) decreases. The image resolution is resized to 64.

**Certainty and diversity.** Indeed, feeding conditional input images (Fig. 18(a)) into the deresidual and denoising network enhances the certainty of the generated images, while diminishing diversity, as shown in Fig. 18(j-l). Generating a clear target image directly from a noisy-carrying degraded image (Fig. 18(b)) without any conditions increases diversity, but changes non-missing regions (Fig. 18(f)).

**Noise Perturbation Intensity.** Table 8 shows that for image restoration, our RDDM with SM-Res or SM-Res-N is not sensitive to the noise intensity $\bar{\beta}_T^2$. For image generation, the diversity of the

Table 8: Experiment with varying $\bar{\beta}_T^2$ for shadow removal on the ISTD dataset (Wang et al., 2018) and low-light enhancement on the SID-RGB dataset (Xu et al., 2020).

| Dataset | Sampling | $\bar{\beta}_T^2$ | PSNR | SSIM |
|---------|----------|-------------------|------|------|
| ISTD | SM-Res-N | 0.01 | **30.91** | **0.962** |
|      |          | 1 | 30.85 | 0.961 |
| SID-RGB | SM-Res | 0.01 | **24.07** | **0.830** |
|         |        | 1 | 23.83 | 0.833 |

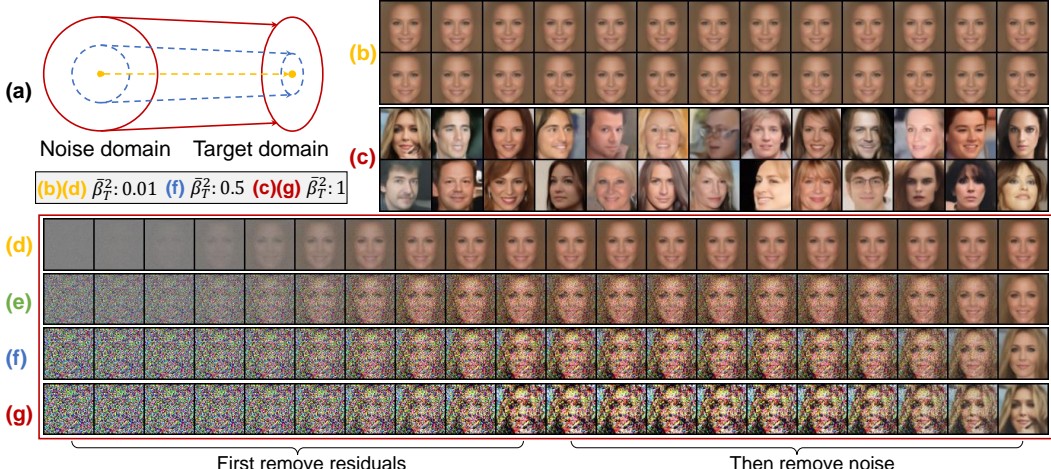

Figure 19: Visualizing ablation studies of noise perturbation intensity ($\bar{\beta}_T^2$). (a) We change the variance ($\bar{\beta}_T^2$) during testing, specifically by coefficient transformation via Algorithm 1. (b-c) When $\bar{\beta}_T^2$ decreases from 1 in (c) to 0.01 in (b), the diversity of the generated images decreases significantly. (d-g) We visualize each step in the generation process. $\bar{\beta}_T^2 = 0.01$ in (d), $\bar{\beta}_T^2 = 0.1$ in (e), $\bar{\beta}_T^2 = 0.5$ in (f), and $\bar{\beta}_T^2 = 1$ in (g). The sampling method is SM-Res-N-2Net with 10 sampling steps.

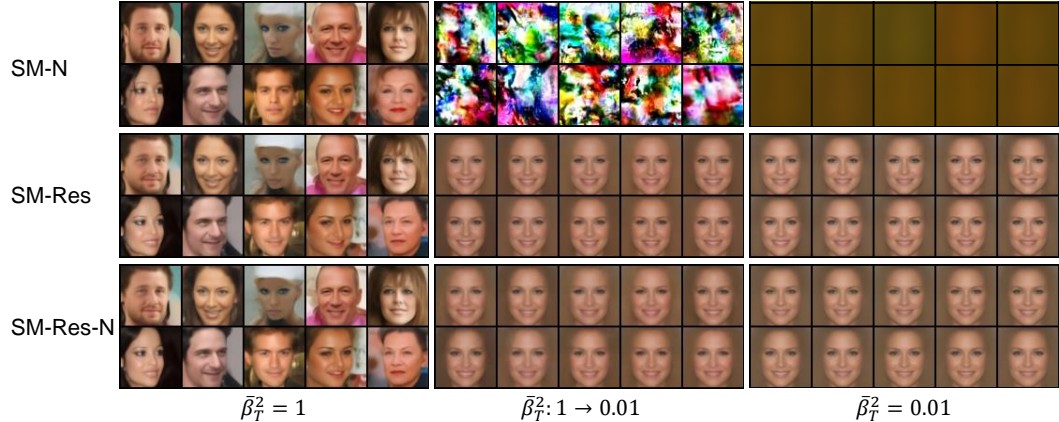

Figure 20: Visualizing ablation studies of sampling methods with different intensities of noise perturbation ($\bar{\beta}_T^2$). "$\bar{\beta}_T^2 : 1 \to 0.01$" denotes that the variance ($\bar{\beta}_T^2$) is changed during testing by coefficient transformation via Algorithm 1. For $\bar{\beta}_T^2 = 1$ and $\bar{\beta}_T^2 = 0.01$, the variance ($\bar{\beta}_T^2$) is the same for training and testing. The sampling steps are 10.

generated images decreases significantly as $\bar{\beta}_T^2$ decreases from 1 in Fig. 19(c) to 0.01 in Fig. 19(b). The experiment is related to the mean face (Wilson et al., 2002; Loffler et al., 2005; Hu & Hu, 2021; Meng et al., 2021b) and could provide useful insights to better understand the generative process. Fig. 20 shows that modifying $\bar{\beta}_T^2$ during testing ($\bar{\beta}_T^2 : 1 \to 0.01$) causes SM-N to fail to generate meaningful faces. SM-Res-N including deresidual and denoising networks can generate meaningful face images like SM-Res, indicating that the denoising network can perform denoising when modifying $\bar{\beta}_T^2$, but cannot obtain robust residuals ($I_{res}^\theta$) in the sampling process by Eq. 12. In summary, the deresidual network is relatively robust to noise variations compared to the denoising network.

**Resource efficiency.** Due to fewer sampling steps, our RDDM inference time and performance is comparable to lllflow (Wang et al., 2022a), and LLFormer (Wang et al., 2023) (not diffusion-based). Compared to SR3 (Saharia et al., 2022), our RDDM (only res in Table 9(b)) has 10x fewer training iterations, 10x fewer parameters, 10x faster inference time, and 10% improvement in PSNR and SSIM

Table 9: Resource efficiency and performance analysis by THOP. "MAC" means multiply-accumulate operation. (a) Low-light enhancement on the LoL dataset (Wei et al., 2018). (b) Shadow removal on the ISTD dataset (Wang et al., 2018). For a fair comparison, a priori shadow mask are used in SR3 with a batch size of 1. (c) Deraining on the RainDrop dataset (Qian et al., 2018).

| (a) Low-light | PSNR(↑) | SSIM(↑) | LPIPS(↓) | Params(M) | MACs(G)×Steps | Inference Time(s) |
|---|---|---|---|---|---|---|
| LLformer | 23.649 | 0.816 | 0.169 | 24.51 | **22.0×1 = 22.0** | 0.09×1 = 0.09 |
| LLFlow | 25.19 | 0.93 | **0.11** | 17.42 | 286.33×1 = 286.3 | 0.18×1 = 0.18 |
| Ours(RDDM) | **25.392** | **0.937** | 0.134 | **7.73** | 32.9×2 = 65.8 | **0.03×2 = 0.06** |

| (b) Shadow Removal | MAE(↓) | PSNR(↑) | SSIM(↑) | Params(M) | MACs(G) × Steps | Inference Time(s) |
|---|---|---|---|---|---|---|
| Shadow Diffusion | **4.12** | **32.33** | **0.969** | - | - | - |
| SR3 Saharia et al. (2022) (80k) | 14.22 | 25.33 | 0.780 | 155.29 | 155.3×100=15530.0 | 0.02×100 = 2.00 |
| SR3 Saharia et al. (2022) (500K) | 13.38 | 26.03 | 0.820 | 155.29 | 155.3×100=15530.0 | 0.02×100 = 2.00 |
| SR3 Saharia et al. (2022) (1000K) | 11.61 | 27.49 | 0.871 | 155.29 | 155.3×100=15530.0 | 0.02×100 = 2.00 |
| Ours (only res, 80k) | 4.76 | 30.72 | 0.959 | **7.74** | **33.5×5 = 167.7** | **0.03×5 = 0.16** |
| Ours (80k) | 4.67 | 30.91 | 0.962 | 15.49 | 67.1×5 = 335.5 | 0.06×5 = 0.32 |

| (c) Deraining | PSNR(↑) | SSIM(↑) | Params(M) | MACs(G) × Steps | Inference Time(s) |
|---|---|---|---|---|---|
| RainDiff64[28] | 32.29 | 0.9422 | 109.68 | 252.4×10 = 2524.2 | 0.03×10 = 0.38 |
| RainDiff128[28] | 32.43 | 0.9334 | 109.68 | 248.4×50 = 12420.0 | 0.038×50 = 1.91 |
| Ours (only res) | 31.96 | 0.9509 | **7.73** | **32.9×5 = 164.7** | **0.032×5 = 0.16** |
| Ours | **32.51** | **0.9563** | 15.47 | 65.8×5 = 329.3 | 0.07×5 = 0.35 |

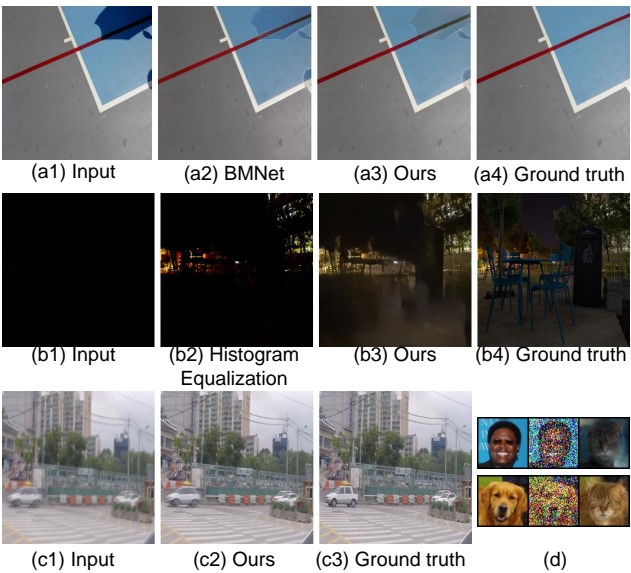

(a1) Input  (a2) BMNet  (a3) Ours  (a4) Ground truth

(b1) Input  (b2) Histogram Equalization  (b3) Ours  (b4) Ground truth

(c1) Input  (c2) Ours  (c3) Ground truth  (d)

Figure 21: Failure cases. (a1-a4) Shadow removal on the ISTD dataset (Wang et al., 2018). (b1-b4) Low-light enhancement on the SID-RGB dataset (Xu et al., 2020). (c1-c3) Deblurring on the GoPro (Nah et al., 2017) dataset. (d) Image translation (male/dog→cat) on the CelebA-HQ (Karras et al., 2018) and AFHQ (Choi et al., 2020) datasets.

on the ISTD (Wang et al., 2018) dataset (shadow removal). For a fair comparison, priori shadow masks are used in SR3 (Saharia et al., 2022) with a batch size of 1. ShadowDiffusion (Guo et al., 2023) uses SR3 (Saharia et al., 2022) and Uformer (Wang et al., 2022b), which has a higher PSNR but is also expected to be more computationally expensive. **Our RDDM with SM-Res requires only 4.8G of GPU memory for training.** Experiments in shadow removal and low-light enhancement demonstrate the effectiveness of RDDM, enabling computationally-constrained researchers to utilize diffusion modeling for image restoration tasks.

**Accelerating Convergence.** The residual prediction in our RDDM helps the diffusion process to be more certain, which can accelerate the convergence process, e.g., fewer training iterations and higher performance in Table 9(b).

**Failure case.** We present some failure cases in Fig. 21.

## C  DISCUSSIONS, LIMITATIONS, AND FURTHER WORK

**Limitations.** Our primary focus has been on developing a unified prototype model for image restoration and generation, which may result in certain performance limitations when compared to task-specific state-of-the-art methods. To further improve the performance of a specific task, potential avenues for exploration include using a UNet with a larger number of parameters, increasing the batch size, conducting more iterations, and implementing more effective training strategies, such as learning rate adjustments customized for different tasks. For the image generation task, although Table 2 showcases the development of an improved coefficient schedule, attaining state-of-the-art performance in image generation necessitates further investigation and experimentation. In summary, while we recognize the existing performance limitations for specific tasks, we are confident that our unified prototype model serves as a robust foundation for image restoration and generation.

**Further Work.** Here are some interesting ways to extend our RDDM.

1. In-depth analysis of the relationship between RDDM and curve/multivariate integration.
2. Development of a diffusion model trained with one set of pre-trained parameters to handle several different tasks.
3. Implementing adaptive learning coefficient schedules to reduce the sampling steps while improving the quality of the generated images.
4. Constructing interpretable multi-dimensional latent diffusion models for multimodal fusion, e.g., generating images using text and images as conditions.
5. Adaptive learning noise intensity ($\beta_T^2$) for an unknown new task.

**Broader Impacts.** Our work establishes a seamless connection between denoising-based diffusion models and image restoration tasks, which serves the broader impact and potential of diffusion models in various fields. By introducing a directional residual diffusion approach with perturbations, our method holds promise for fine-tuning generation tasks. For instance, it can be applied to fine-tune classic clothing styles, model temporal variations with perturbations during face sculpting, or simulate realistic device aging processes. Furthermore, deterministic implicit sampling techniques (similar to DDIM (Song et al., 2021a) and GANs) can be employed to explore data compression and encryption.

Nevertheless, it is important to acknowledge the potential misuse and ethical concerns associated with data generation and encryption. For example, fake image videos of high-profile individuals raise ethical concerns, as they can be exploited to deceive, manipulate, or spread false information. Additionally, when it comes to encrypted data, our approach poses regulatory challenges. Encryption plays a crucial role in safeguarding sensitive information, but it can also hinder regulatory efforts aimed at combating illicit activities or ensuring public safety.

