# OpenReview forum: "Residual Denoising Diffusion Models"
_ICLR.cc/2024/Conference — ICLR 2024 Conference Withdrawn Submission_

### Official Review · Reviewer_TatZ · 2023-10-17

**Soundness:** 2 fair
**Presentation:** 2 fair
**Contribution:** 2 fair
**Rating:** 5
**Confidence:** 4

**Summary:**

This paper proposes residual denoising diffusion models (RDDM), which decouples the traditional single denoising diffusion process into
residual diffusion and noise diffusion.

**Strengths:**

This paper compares various tasks.

**Weaknesses:**

*The data provided for inpainting tasks is limited. I would like the authors to conduct quantitative comparisons, especially on the CelebA and Places2 datasets, and compare with state-of-the-art diffusion models (2,3), including considerations of efficiency and parameter count.

*Residual diffusion has been explored and discussed extensively in the domain of image restoration (1,2). It should be discussed in detail and compared with them.

*Please provide a comparison of the computational complexity, runtime, and parameter count for the methods being compared.

*For the deraining task in Table 3, please compare it with Restormer using several standard datasets, including comparisons of runtime efficiency and parameter count.


(1) Srdiff: Single image super-resolution with diffusion probabilistic models
(2) Diffir: Efficient diffusion model for image restoration
(3) Repaint: Inpainting using denoising diffusion probabilistic models
(4) Restormer: Efficient transformer for high-resolution image restoration

**Questions:**

check weakness

---

### Official Review · Reviewer_a71e · 2023-10-17

**Soundness:** 3 good
**Presentation:** 2 fair
**Contribution:** 3 good
**Rating:** 6
**Confidence:** 4

**Summary:**

The paper introduces Residual Denoising Diffusion Models (RDDM), a dual diffusion process. The proposed RDDM decouples the diffusion process into residual diffusion and noise diffusion, which can unify the image restoration and generation process. The authors demonstrate the consistency of RDDM with the diffusion models DDPM and DDIM by transforming coefficient schedules. Additionally, they propose a partially path-independent generation process. Experiments demonstrate the effectiveness of RDDM.

**Strengths:**

1. The authors propose RDDM as a unified framework for image restoration and generation. It offers a versatile approach to these related tasks.
2. The authors demonstrate the consistency of RDDM with DDPM/DDIM through coefficient schedule transformations.
3. The proposed partially path-independent generation process decouples residuals and noise, and reasonably explains the role of the two branches.
4. They provide the code, which shows the solidness of the work.

**Weaknesses:**

The paper lacks perceptual metrics or a detailed comparison for some tasks, such as Low-light and Deraining, where RDDM performs better regarding PSNR and SSIM. It would be beneficial to provide more comprehensive comparisons, especially using perceptual metrics. Additionally, the paper evaluates the performance at 5 steps for shadow removal and deraining, and 2 steps for low-light and deblurring. Since step numbers affect performance, it is recommended to analyze the impact of different numbers of steps on performance.

**Questions:**

1. In Table 1, RDDM performs better in different strategies for two restoration tasks, Low-light (LOL) and Deraining (RainDrop). It would be beneficial to explain this phenomenon.
2. Provide more comparisons in terms of different metrics and evaluate the model's performance with more steps in Low-light and Deraining.

---

### Official Review · Reviewer_JomK · 2023-10-28

**Soundness:** 3 good
**Presentation:** 3 good
**Contribution:** 2 fair
**Rating:** 5
**Confidence:** 5

**Summary:**

This paper proposes a unified model for image generation and restoration with the concept of residual. The proposed method is compatible with different diffusion models and sampling strategies. It can also extend to image restoration task.

**Strengths:**

1. This paper unified image generation and generation in a framework.
2. This paper demonstrates its effectiveness on various image restoration task.
2. This paper is well-writen and easy-to-understand.

**Weaknesses:**

1. The novelty is limited. It seems more like a combination of two existing components. The concept of residual is not rare in diffusion models. Several previous works employ the same idea in image restoration task [1][2]. Method [1] already introduced the residual concept in restoration task and verified the effectiveness. Beseds, this paper didn't cite these methods.
2. This method only mathetically combines image generation and restoration. In the experiments, it employ pretrained diffusion models for image generation with only coefficient transformation. While for image restoration, it needs to retrain the diffusion model for different restoration tasks.
[1] Image restoration with mean-reverting stochastic differential equations
[2] Resshift: Efficient diffusion model for image super-resolution by residual shifting.

**Questions:**

Refer to the weakness and questions mentioned above.

---

### Official Review · Reviewer_vFvZ · 2023-11-01

**Soundness:** 3 good
**Presentation:** 3 good
**Contribution:** 3 good
**Rating:** 5
**Confidence:** 4

**Summary:**

The authors propose a novel framework for diffusion models named residual denoising diffusion models (RDDM), which is used for image restoration and image generation. RDDM decouples the diffusion framework into residual diffusion and noise diffusion. The former represents directed diffusion and the latter represents randomness in the diffusion process.  Qualitative and quantitative experiments showed the superiority of the method.

**Strengths:**

1. The results in the paper show significant improvement in image generation and image restoration.
2. The proposed architectural provides a new idea for the interpretability of diffusion models and more accurate results can be obtained with fewer sampling steps.

**Weaknesses:**

1. Sampling speed in reverse process is an important factor that influences the quality of diffusion models. Compared with DDPM, DDIM, and other diffusion models, does RDDM have an advantage in sampling speed?

2. In Eq.10 (reverse process), the author sets \eta to 0, which represent a deterministic generation process. What is the advantage to set \eta to 0?

3. The author's approach to minimizing the loss function is minimizing the upper bound of Eq. 25, i.e. minimizing Eq. 26. However, it seems that in the derivation from Equation 25 to Eq.  26, the author did not explicitly consider the relationship of \alpha_t and \frac{\beta_t^2}{\overline{\beta}} between Eq.25 and Eq.26. And the author only utilizes \lambda_{res} and \lambda_{\varepsilon} as coefficients for the loss function. How can author ensure that Eq. 26 is indeed an upper bound for Eq. 25?

**Questions:**

Please refer to weakness.